# Divide and Contrast:
# Learning Robust Temporal Features without Augmentation

Abdul-Kazeem Shamba [1]   Kerstin Bach [1]   Gavin Taylor [2]

## Abstract

Self-supervised learning for time-series representation aims to reduce reliance on labeled data while maintaining strong downstream performance, yet many existing approaches incur high computational costs or rely on assumptions that do not hold across diverse temporal dynamics. In this work, we introduce Divide and Contrast (Di-COT), an unsupervised framework that avoids data augmentation and multiple encoder passes by contrasting informative substructures within a window rather than individual timesteps. Di-COT stochastically partitions each window into a small number of overlapping sub-blocks per iteration, enabling efficient and meaningful contrast while mitigating false positives during temporal transitions. To further improve scalability, we adopt a contrastive objective whose computation depends on the batch size and the number of sub-blocks, making loss computation independent of sequence length. Extensive experiments on six large-scale real-world datasets, as well as the UCR and UEA benchmarks, demonstrate that Di-COT learns semantically structured and transferable representations, achieving state-of-the-art performance on classification, clustering, $k$NN, and cross-dataset transfer, while substantially reducing training time. The source code is publicly available at https://github.com/sfi-norwai/Di-COT.

## 1. Introduction

Traditional supervised learning is expensive because it relies on large labeled datasets and domain-specific annota-

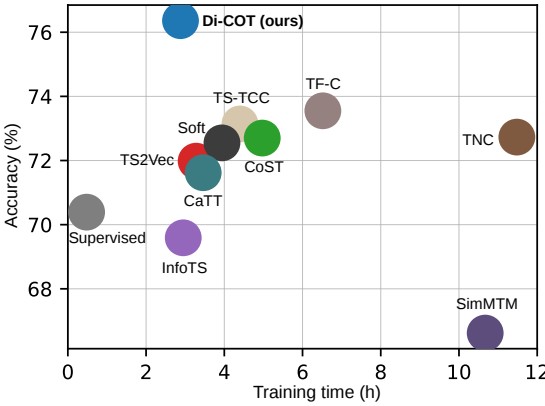

*Figure 1.* **Average accuracy vs. training time comparison.** Accuracy is averaged over five seeds and six large-scale datasets ($> 20$k samples). Training time denotes the cumulative training time across all six datasets. For SSL methods, encoders are pretrained on the full training set and evaluated with a frozen backbone and a linear probe trained on 1% of the training data, while the supervised baseline is trained end-to-end on 1% labeled data.

tions to achieve good performance. Self-supervised learning learns directly from the data, eliminating the need for annotation, to improve downstream performance on limited datasets. Contrastive learning has emerged as a promising technique for self-supervised representation learning (Oord et al., 2018; Chen et al., 2020). Given the success of these techniques in computer vision (CV) and natural language processing (NLP) over the past decade, researchers have sought to employ them for self-supervised representation learning (SSL) in time series to improve performance in downstream tasks (Tonekaboni et al., 2021; Woo et al., 2022; Yue et al., 2022; Luo et al., 2023; Foumani et al., 2024; Duan et al., 2024). However, these techniques have speed limitations due to the sampling strategies and the loss computations. Specifically, these methods either use augmentation and masking for positive pair selection, apply dynamic time warping (DTW) to compute similarities, or involve multiple passes through the encoder for the positives, negatives, and anchors. As a consequence, either the speed is affected, or the augmentation introduces bias in the learning that hinders the generalizability of the learned feature embeddings. An additional method, CaTT (Shamba et al., 2025), avoided some of these disadvantages by adopting a

[1]Department of Computer Science, Norwegian University of Science and Technology, Norway [2]Department of Computer Science, United States Naval Academy, USA. Correspondence to: Abdu-Kazeem Shamba <abdul.k.shamba@ntnu.no>.

*Proceedings of the 43rd International Conference on Machine Learning*, Seoul, South Korea. PMLR 306, 2026. Copyright 2026 by the author(s).

sampling strategy that contrasts every timestep within an instance. However, this approach relies on an assumption of temporal smoothness between consecutive timesteps, which may not hold for many datasets in the UCR (Dau et al., 2019) and UEA (Bagnall et al., 2018) benchmarks, where sequence lengths and temporal dynamics vary substantially, leading to degraded performance.

To address these limitations, we introduce Divide and Contrast (Di-COT). Di-COT learns representations by stochastically partitioning each instance into a variable number of overlapping sub-blocks sampled from a uniform distribution. The method treats temporally adjacent sub-blocks as positive pairs while designating all other segments in the same sequence as negatives. By contrasting only within-instance substructures, Di-COT ensures that positives share the same semantic context, avoiding false positives across temporal transitions. Furthermore, this sub-block approach prevents the model from contrasting trivial timesteps and makes the loss computation independent of sequence length. Extensive experiments on six large real-world datasets (>20k samples) as well as on the smaller UCR and UEA datasets demonstrate that Di-COT learns generalizable and transferable features while maintaining the lowest training time. Our contributions can be summarized below:

- We propose a framework that eliminates the need for data augmentation, timestep masking, and multiple encoder passes, thereby reducing computational overhead and preventing representation distortion.

- We introduce dynamic sub-block contrasting within individual instances, preventing trivial timestep-level discrimination while promoting diverse and semantically meaningful temporal representations across training.

- We reformulate temporal contrastive learning as a cross-entropy classification task, which provides dense supervision by utilizing every sub-block as a training signal while retaining the properties of the standard temperature-scaled InfoNCE objective.

- We demonstrate state-of-the-art performance and training efficiency across six large-scale time-series datasets and the UCR/UEA benchmarks.

## 2. Related Works

**Contrastive representation learning.** Contrastive learning (CL) (Hadsell et al., 2006) has become a foundational paradigm for self-supervised representation learning, particularly in computer vision and natural language processing. Rather than reconstructing inputs, contrastive methods learn representations by pulling together semantically related samples while pushing apart unrelated ones. Early formulations such as the triplet loss (Schroff et al., 2015) rely on explicitly sampled anchor–positive–negative tuples, which can be inefficient and sensitive to sampling quality. The N-pair loss (Sohn, 2016) improves sample efficiency by jointly contrasting a positive against multiple negatives within a batch, enabling more stable optimization. Contrastive Predictive Coding (CPC) (Oord et al., 2018) extends this idea to sequential data by predicting future latent representations using an autoregressive model and the InfoNCE objective, derived from noise-contrastive estimation (Gutmann & Hyvärinen, 2010). SimCLR (Chen et al., 2020) demonstrates that strong data augmentation combined with a normalized temperature-scaled contrastive loss (NT-Xent) can yield high-quality representations, where all other samples in the batch implicitly act as negatives. Momentum-based methods such as MoCo (He et al., 2020) further stabilize training by maintaining a slowly updated encoder and a dynamic feature dictionary. Subsequent work refines contrastive objectives and positive pair selection, including regularization-based invariance (Mitrovic et al., 2021), modifications to the InfoNCE denominator (Yeh et al., 2022), and nearest-neighbor–based positives (Dwibedi et al., 2021). Despite their success, these approaches are largely developed for static data and often depend on heavy data augmentation, large batch sizes, or multiple encoder passes, which limits their efficiency and applicability to long, structured temporal signals.

**Contrastive learning for time series.** Motivated by advances in CL, recent work has adapted contrastive objectives to time-series (TS) representation learning by designing task-specific sampling strategies for positive and negative pairs. Many methods rely on augmentation to generate contrasting views. For example, TF-C (Zhang et al., 2022) enforces consistency between time-domain and frequency-domain representations, while TimeCLR (Yang et al., 2022) introduces DTW-based augmentations to capture temporal distortions such as phase shifts and amplitude variations. Other approaches incorporate temporal structure more explicitly. TS2Vec (Yue et al., 2022) defines positive pairs as representations at the same timestamp across different views and jointly optimizes instance-wise and temporal contrastive objectives. InfoTS (Luo et al., 2023) studies principled criteria for selecting effective augmentations in the TS domain. T-loss (Franceschi et al., 2019) and TNC (Tonekaboni et al., 2021) select positives and negatives based on temporal proximity, using either fixed distance rules or statistically defined neighborhoods. Soft (Lee et al., 2024) introduces soft assignments over instance and temporal contrastive pairs, while CLOCs (Kiyasseh et al., 2021) defines positives as transformed segments originating from the same subject. TS-TCC (Eldele et al., 2021) integrates temporal and contextual cues by predicting future representations of one augmented view from the past features of another. In forecasting-oriented settings, CoST (Woo et al., 2022) embeds inductive biases to disentangle seasonal components, and MF-CLR (Duan et al., 2024) further explores hierarchical modeling of multi-frequency information. While

effective, many of these methods suffer from practical limitations. Augmentation and masking can introduce distortions or inductive biases, DTW-based similarity computation is expensive for long sequences, and several approaches require multiple encoder passes for anchors, positives, and negatives. Moreover, sampling strategies that contrast only a small number of temporal pairs per update underutilize available data and reduce training efficiency.

**Temporal adjacency and efficiency-driven contrastive learning.** Our work is most similar to more recent work, CaTT (Shamba et al., 2025). CaTT eliminates augmentation and multiple encoder passes by contrasting temporally adjacent representations within a sequence, treating nearby timestamps as positives and leveraging in-batch negatives. This significantly improved training speed and scalability and showed strong performance on long real-world time series. However, this relies on the implicit assumption that temporal adjacency implies semantic similarity, which does not hold in datasets where event change frequently, which are present in the UCR and UEA benchmarks. As a result, contrasting all consecutive timestamps across multiple instances can introduce false positives during transitions and encourage trivial or semantically meaningless contrasts. In addition, contrasting every timestep within an instance and batch leads to similarity matrices whose computation and memory cost scale with both sequence length and batch size, limiting efficiency for long windows. These limitations highlight a key trade-off in existing TS contrastive learning methods: approaches that aggressively exploit temporal proximity gain efficiency but struggle to generalize when temporal continuity does not align with semantic consistency. Our method addresses this challenge by limiting contrast to informative substructures within an instance, and adopting a contrastive objective whose computation is independent of sequence length, enabling both efficient training and improved generalization.

# 3. Divide and Contrast (Di-COT)

Given a batch of $B$ time-series instances, each of length $T$ with input dimensionality $D$, Di-COT learns a temporal encoder $f_\theta : \boldsymbol{x} \rightarrow \boldsymbol{z}$, that maps an input sequence to a sequence of latent representations. Formally, let

$$\boldsymbol{x} = \{x^{(1)}, x^{(2)}, \ldots, x^{(B)}\}, \quad x^{(i)} \in \mathbb{R}^{T \times D},$$

where $x^{(i)}$ denotes the $i$-th instance in the batch. The encoder produces $\boldsymbol{z}^{(i)} = f_\theta(x^{(i)})$, $z^{(i)} \in \mathbb{R}^F$, where $F$ denotes the embedding dimension.

## 3.1. Divide and Contrast Pretext Task

To enable self-supervised contrastive learning without augmentations, during **training** each sequence is stochastically partitioned per-iteration into $k$ overlapping sub-blocks

$k \sim \mathcal{U}\{k_{\min}, \ldots, k_{\max}\}$ of length $L = \frac{T}{1+(k-1)(1-\rho)}$ and stride $s = \lfloor L(1-\rho) \rceil$, with overlap ratio $\rho \in (0,1)$:

$$\tilde{x}^{(i)} = \{\tilde{x}_1^{(i)}, \ldots, \tilde{x}_k^{(i)}\}, \quad \tilde{x}_j^{(i)} \in \mathbb{R}^{L \times D}.$$

Each sub-block is encoded and pooled along the temporal dimension to produce a single embedding:

$$z_j^{(i)} = f_\theta(\tilde{x}_j^{(i)}) \in \mathbb{R}^F,$$

so that the $i$-th instance is represented by its sub-block embeddings $\mathbf{z}^{(i)} = \{z_1^{(i)}, \ldots, z_k^{(i)}\}$.

Temporally adjacent sub-blocks are treated as positive pairs, and all other sub-blocks in the same sequence are negatives. This yields a dense, in-sequence contrastive supervision without augmentations or multiple encoder passes. Moreover, because the number of sub-blocks $k$ is randomly sampled at each iteration, the encoder is exposed to multiple temporal granularities during training, implicitly encouraging representations that are robust across both short and long-range temporal contexts and enabling the capture of multi-scale dynamics without an explicit hierarchical or multi-resolution design.

## 3.2. Contrastive Objective

For each instance $i \in \{1, \ldots, B\}$, we compute a temperature-scaled similarity matrix

$$\mathbf{S}^{(i)} \in \mathbb{R}^{k \times k}, \quad S_{j,p}^{(i)} = \frac{z_j^{(i)\top} z_p^{(i)}}{\tau},$$

where $\tau$ is a temperature hyperparameter. Each row $\mathbf{S}_{j,:}^{(i)}$ defines a categorical distribution over all sub-block positions.

Di-COT formulates temporal contrastive learning as a multi-class classification problem (Algorithm 1). For each sub-block $j$, the positive label corresponds to its true preceding temporal neighbor. For the first sub-block, where no preceding position exists, the target index is set to zero:

$$p^*(j) = \begin{cases} 0, & j = 0, \\ j-1, & j > 0. \end{cases}$$

The training objective is defined as:

$$\mathcal{L}_{\text{CE}} = -\frac{1}{Bk} \sum_{i=1}^{B} \sum_{j=0}^{k-1} \log \frac{\exp\left(S_{j,p^*(j)}^{(i)}\right)}{\sum_{p=0}^{k-1} \exp\left(S_{j,p}^{(i)}\right)}. \quad (1)$$

Rather than predicting raw values or exact future sub-blocks, Di-COT encourages embeddings of temporally adjacent sub-blocks to be similar in representation space, while contrasting them against all other sub-blocks in the sequence. This

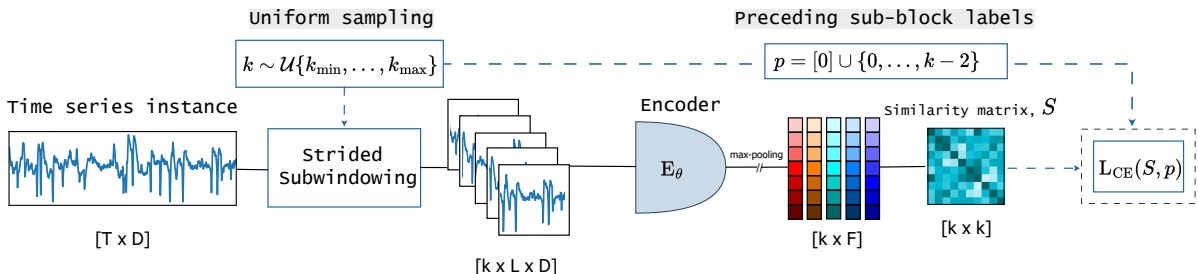

*Figure 2.* **Di-COT framework.** Each time-series window is stochastically divided into $k$ overlapping sub-blocks, which are encoded and contrasted via a next-sub-block predictive objective based on their pairwise similarities.

maximizes translational robustness, ensuring that even small shifts in the instance window produce similar embeddings. In this sense, Di-COT replaces value-space prediction with representation-space discrimination, focusing on learning relational structure rather than generative modeling.

Under this formulation, Di-COT's loss computation achieves a time complexity of $\mathcal{O}(Bk^2d)$ and a memory complexity of $\mathcal{O}(Bk^2)$, where $B$ is the batch size, $k$ is the number of sub-blocks, and $d$ is the embedding dimension.

Since $k \ll T$ by design, this ensures that the method is efficient even for very long instances (Appendix C).

This formulation can be interpreted as a *temporal contrastive classification* task in representation space, where each sub-block serves as a training signal and similarity scores against all other sub-blocks are treated as logits. Unlike pairwise contrastive objectives such as in Tonekaboni et al. (2021) and Yue et al. (2022), Di-COT provides dense supervision by leveraging every sub-block as an anchor, resulting in $B \times k$ positive assignments per update compared to $2B$ in augmentation-based contrastive methods. By directly optimizing a stable cross-entropy objective, Di-COT achieves improved memory efficiency and significantly faster training on long sequences, while preserving the core properties of temperature-scaled InfoNCE (Appendix A).

## 4. Experiments

To comprehensively evaluate the robustness and generality of Di-COT, we conduct extensive experiments across a diverse set of downstream tasks, including *linear probing*, *low-label regimes*, *kNN evaluation*, *clustering*, and *cross-dataset transfer*. We perform experiments on six publicly available large-scale real-world time-series datasets with more than 20,000 samples: PAMAP2 (Reiss & Stricker, 2012), WISDM2 (Weiss & Lockhart, 2012), HARTH (Logacjov et al., 2021), SLEEP (Goldberger et al., 2000), ECG (Moody & Mark, 1983; Goldberger et al., 2000), and SKODA (Zappi et al., 2012). These datasets span a wide range of temporal characteristics, with sequence lengths varying from $T = 40$ to $T = 3000$, and input dimension-

**Algorithm 1** Divide and Contrast (Di-COT)

**Input:** Batch of time series $X \in \mathbb{R}^{B \times T \times D}$
**Parameters:** Encoder $f_\theta$, overlap ratio $\rho$, sub-block range $[k_{\min}, k_{\max}]$, temperature $\tau$
- Sample number of sub-blocks, $k \sim \mathcal{U}\{k_{\min}, \ldots, k_{\max}\}$
- Create overlapping sub-blocks using strided view $X_{\text{sub}} \in \mathbb{R}^{B \times k \times L \times D}$, then
- Reshape $X_{\text{sub}} \leftarrow \mathbb{R}^{(B \cdot k) \times L \times D}$
- Encode all sub-blocks, $Z = f_\theta(X_{\text{sub}}) \in \mathbb{R}^{(B \cdot k) \times F}$
- Compute similarity, $S_{B,k,k} = \frac{1}{\tau} \sum_{f=1}^F Z_{B,k,f}^{(1)} \cdot Z_{B,k,f}^{(2)}$
- Assign ordered labels $p = [0] \cup \{0, \ldots, k-2\}$ for preceding sub-block prediction. Repeat for batch size $B$
- Flatten similarity matrix and labels $S \leftarrow \mathbb{R}^{(B \cdot k) \times k}$ & $p \leftarrow \mathbb{R}^{(B \cdot k)}$
**Output:** Compute cross-entropy loss: $\mathcal{L} = \text{CE}(S, p)$

alities ranging from $D = 1$ to $D = 64$ (Appendix F). To further demonstrate that Di-COT is also effective for smaller datasets with frequent event changes, we additionally evaluate on both the univariate UCR and the multivariate UEA archives. These benchmarks cover a broad spectrum of data distributions and application domains, making them a rigorous testbed for time-series classification.

**Baselines.** We compare Di-COT against a diverse set of strong self-supervised learning methods for time series, including several state-of-the-art (SOTA) approaches: TNC (Tonekaboni et al., 2021), TS-TCC (Eldele et al., 2021), TS2Vec (Yue et al., 2022), CoST (Woo et al., 2022), TF-C (Zhang et al., 2022), InfoTS (Luo et al., 2023), SimMTM (Dong et al., 2023), Soft (Lee et al., 2024), and CaTT (Shamba et al., 2025). In addition, we include traditional non-deep-learning baselines such as MiniRocket (Dempster et al., 2021) and Random Forest (RF)(Breiman, 2001), as well as linear classifiers trained on raw inputs and randomly initialized encoders (More details on Appendix E).

**Training Details.** To isolate the impact of the learning framework from architectural choices, all methods are trained using the same InceptionTime backbone (Ismail Fawaz et al., 2020), with a unified data-loading pipeline

and aligned hyperparameters wherever applicable. All models are optimized using AdamW with a learning rate of $3 \times 10^{-4}$, weight decay of $3 \times 10^{-4}$, and momentum parameters $\beta_1 = 0.9$, $\beta_2 = 0.99$. We employ a cosine learning-rate schedule with linear warm-up during the first $10\%$ of training iterations. For large-scale datasets, we train all models for 1500 iterations using a batch size of 128. Due to memory constraints imposed by long sequence lengths, the batch size is reduced to 16 for the SLEEP and ECG datasets. For the UEA and UCR benchmarks, we follow the training protocol in Yue et al. (2022): models are trained for 600 iterations on datasets with effective size greater than 100,000, and for 200 iterations otherwise, where effective size is defined as the product of the number of samples and the window length. A batch size of 8 is used for all UEA and UCR experiments. Across all experiments, representations are projected to a fixed embedding dimension of $d = 256$. All experiments are conducted on NVIDIA V100 GPUs.

## 4.1. Linear Evaluation with Frozen Backbone

Linear evaluation aims to assess the usefulness of the learned representations for downstream classification tasks. To this end, we freeze the pretrained backbone and train a logistic regression classifier on the representations extracted from the backbone encoder. Backbones are trained from a number of previous methods, and results are compared for both accuracy and speed. In addition, we compare against a number of other approaches. A fully supervised model trained end-to-end serves as an upper bound, and fitting the logistic regressor to the raw features serves as a lower bound. We also train a random forest (RF) classifier and MiniRocket, to contrast against other plausible machine learning approaches. We first discuss results from these experiments on large datasets; Table 1 reports these results.

In terms of average accuracy, Di-COT outperforms all methods except the fully supervised end-to-end model, achieving the highest accuracy on three of the six datasets (ECG, HARTH, and SKODA), and second-best on two more. Crucially, Di-COT achieves this while having the lowest training time among DL-based approaches. Since all methods share the same backbone and training configuration, FLOPs are identical; we therefore report wall-clock training time to capture algorithmic efficiency.

CaTT ranks second overall in average accuracy and has the third-lowest training time. Unlike Di-COT—which splits each sequence into $k$ sub-blocks with $k \ll T$, reducing the similarity matrix to $B \times k \times k$—CaTT retains full temporal resolution $BT \times BT$, making it slower on long-sequence datasets such as SLEEP and ECG. TS2Vec achieves the second-lowest training time, but this efficiency stems from a sampling strategy that discards many time steps, leading to suboptimal representation quality. In contrast, Di-COT

leverages all time steps via sub-block construction, resulting in a strategy that is both computationally efficient and memory-friendly.

The traditional ML baselines exhibit distinct performance characteristics tied to sequence length and dimensionality. RF outperforms all other methods on PAMAP2 (short sequence, high dimensionality) but performs poorly on ECG and SLEEP, suggesting that traditional RF models struggle with long sequence datasets. Similarly, MiniRocket shows competitive performance on WISDM2 (short sequences, few channels) but performs poorly on PAMAP2 and SKODA, indicating struggles with high-dimensional inputs.

Finally, it is noteworthy that a randomly initialized backbone consistently outperforms both RF and MiniRocket features. The fact that a randomly initialized encoder with a 256-dimensional output surpasses MiniRocket's $10,000$-dimensional feature representation highlights that projecting to arbitrarily high-dimensional spaces does not necessarily yield better performance. Instead, meaningful inductive structure in the representation is critical, and even an untrained InceptionTime backbone produces more informative features than fixed, handcrafted feature extractors.

We also perform these experiments with the UCR and UEA archives, which serve as a stringent stress test due to their small training sets and heterogeneous domains. The limited data scale challenges SSL methods that typically require large pretraining corpora (Foumani et al., 2024). We therefore evaluate on these archives to assess representation robustness under these constraints. We evaluate all methods on 124 UCR and 28 UEA datasets using the provided train-test splits. We report average classification accuracy, average rank, and total training time aggregated across datasets (Table 2). Across both benchmarks, Di-COT achieves the highest average accuracy, lowest rank, and fastest training time. Figure 3 presents the Critical Difference diagram (Demšar, 2006) based on the Nemenyi test conducted across all datasets, including 6 large-scale datasets, 124 UCR datasets, and 28 UEA datasets. Methods that are not connected by a bold line differ significantly in their average ranks. Our results show that while absolute accuracy gains over CoST and TNC are modest on these small datasets, they are consistent and distinct across a large number of datasets (see Appendix J) and are achieved at a fraction of the training time (0.84 hours vs. 2.48–3.30 hours), indicating a more favorable accuracy–compute trade-off.

Overall, we show that Di-COT produces more useful embeddings, in less time, across a very diverse set of datasets.

## 4.2. Low-Label Regime

The primary objective of self-supervised representation learning is to acquire useful feature representations from

*Table 1.* Linear evaluation accuracy (%) across six large-scale datasets, averaged over five random seeds. We additionally report cumulative training time across all six datasets and average accuracy with 95% confidence intervals. Best in bold, second best underlined.

|  | ECG | HARTH | PAMAP2 | SKODA | SLEEP | WISDM2 | Time (hrs) | Avg. Acc ± 95% CI |
|---|---|---|---|---|---|---|---|---|
| Supervised | $95.33 \pm 1.26$ | $92.33 \pm 2.79$ | $76.10 \pm 0.92$ | $99.90 \pm 0.02$ | $84.99 \pm 0.15$ | $62.81 \pm 5.15$ | - | $85.25 \pm 0.62$ |
| Raw Data | $58.61 \pm 0.00$ | $66.42 \pm 0.00$ | $64.04 \pm 0.00$ | $98.28 \pm 0.00$ | $35.58 \pm 0.00$ | $45.23 \pm 0.00$ | - | $61.36 \pm 0.00$ |
| Random Forest | $53.33 \pm 0.00$ | $79.45 \pm 0.00$ | $\mathbf{72.87 \pm 0.00}$ | $97.57 \pm 0.00$ | $25.27 \pm 0.00$ | $38.47 \pm 0.00$ | - | $61.16 \pm 0.00$ |
| MiniRocket | $68.77 \pm 0.15$ | $44.90 \pm 0.75$ | $10.28 \pm 0.40$ | $24.65 \pm 0.67$ | $59.80 \pm 0.15$ | $\mathbf{66.44 \pm 0.60}$ | - | $46.01 \pm 0.32$ |
| Random Init. | $68.78 \pm 6.16$ | $89.78 \pm 1.32$ | $\underline{71.73 \pm 1.75}$ | $\underline{99.11 \pm 0.12}$ | $82.22 \pm 0.66$ | $59.97 \pm 1.42$ | - | $78.60 \pm 1.27$ |
| TNC | $75.56 \pm 5.28$ | $91.70 \pm 1.81$ | $71.59 \pm 2.90$ | $99.00 \pm 0.12$ | $85.14 \pm 0.30$ | $61.16 \pm 1.87$ | 11.48 | $80.69 \pm 1.28$ |
| TS-TCC | $75.28 \pm 2.32$ | $89.29 \pm 2.17$ | $70.42 \pm 1.40$ | $98.01 \pm 0.29$ | $85.15 \pm 0.34$ | $61.70 \pm 2.58$ | 4.40 | $79.98 \pm 0.72$ |
| TS2Vec | $71.83 \pm 4.14$ | $90.27 \pm 2.00$ | $70.37 \pm 0.46$ | $98.96 \pm 0.05$ | $84.81 \pm 0.21$ | $62.39 \pm 2.69$ | 3.28 | $79.77 \pm 0.88$ |
| TF-C | $74.67 \pm 3.47$ | $92.24 \pm 1.10$ | $71.30 \pm 1.77$ | $98.23 \pm 0.21$ | $85.18 \pm 0.29$ | $62.54 \pm 3.99$ | 6.52 | $80.69 \pm 0.95$ |
| CoST | $73.72 \pm 9.44$ | $88.76 \pm 4.19$ | $71.51 \pm 1.81$ | $98.15 \pm 0.17$ | $84.96 \pm 0.48$ | $60.17 \pm 2.30$ | 4.97 | $79.55 \pm 1.69$ |
| InfoTS | $72.28 \pm 3.31$ | $85.01 \pm 2.15$ | $70.63 \pm 2.25$ | $98.86 \pm 0.14$ | $82.63 \pm 0.87$ | $62.10 \pm 1.30$ | $\underline{2.95}$ | $78.58 \pm 0.65$ |
| SimMTM | $71.33 \pm 3.61$ | $79.23 \pm 5.12$ | $69.75 \pm 0.75$ | $97.60 \pm 0.36$ | $\mathbf{85.22 \pm 0.21}$ | $62.39 \pm 3.72$ | 10.67 | $77.92 \pm 1.25$ |
| Soft | $73.94 \pm 3.76$ | $90.13 \pm 5.18$ | $71.38 \pm 1.86$ | $98.82 \pm 0.17$ | $84.85 \pm 0.25$ | $62.29 \pm 2.96$ | 3.94 | $80.24 \pm 1.44$ |
| CaTT | $\underline{80.89 \pm 1.83}$ | $\underline{93.13 \pm 0.53}$ | $69.86 \pm 0.33$ | $94.87 \pm 0.69$ | $85.17 \pm 0.33$ | $63.25 \pm 3.36$ | 3.47 | $\underline{81.20 \pm 0.41}$ |
| Di-COT | $\mathbf{85.28 \pm 3.72}$ | $\mathbf{93.23 \pm 0.79}$ | $71.38 \pm 0.55$ | $\mathbf{99.41 \pm 0.06}$ | $\underline{85.21 \pm 0.30}$ | $63.92 \pm 2.90$ | $\mathbf{2.88}$ | $\mathbf{83.07 \pm 0.69}$ |

*Table 2.* Classification performance over UCR and UEA datasets, showing average accuracy, average rank, and total training time (hours). Bold indicates the top method per dataset group.

| Method | 124 UCR Datasets | | 28 UEA Datasets | | Total Time (hrs) ↓ |
|---|---|---|---|---|---|
|  | Avg. Acc (%) | Avg. Rank | Avg. Acc (%) | Avg. Rank | (UCR + UEA) |
| Di-COT | **81.33** | **3.73** | **71.24** | **4.64** | **0.84** |
| CoST | 80.03 | 4.65 | 70.02 | 5.00 | 2.48 |
| TNC | 79.63 | 4.70 | 70.38 | 5.32 | 3.30 |
| TF-C | 79.19 | 5.37 | 70.23 | 5.07 | 2.99 |
| TS-TCC | 79.15 | 5.41 | 69.49 | 6.30 | 2.73 |
| Soft | 79.09 | 5.38 | 70.12 | 5.18 | 1.48 |
| CaTT | 77.42 | 6.08 | 69.17 | 6.52 | 0.96 |
| TS2Vec | 77.74 | 6.31 | 69.87 | 5.84 | 2.15 |
| SimMTM | 73.27 | 7.90 | 66.95 | 7.21 | 2.96 |
| InfoTS | 71.96 | 8.53 | 66.87 | 7.09 | 1.66 |
| MiniRocket | 67.89 | 8.18 | 34.43 | 10.87 | – |
| Random Forest | 34.50 | 11.76 | 47.55 | 8.95 | – |

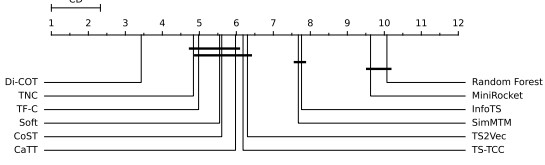

*Figure 3.* Critical Difference (CD) diagram of SSL methods across all dataset categories with a confidence level of 95%.

entirely unlabeled data, thereby reducing annotation costs and reliance on domain expertise. A key indicator of success is strong downstream performance when only a small fraction of labeled data is available, ideally surpassing a fully supervised model trained end-to-end under the same low-label constraints. To evaluate this setting, we follow the same pretraining protocol as in the previous section: all self-supervised learning (SSL) methods are first trained on the full unlabeled training set. Subsequently, we freeze the pretrained backbone and train a logistic regression (LR) classifier using only a small labeled subset. Specifically, we sample a labeled set $\mathcal{D}_\ell \subset \mathcal{D}_{\text{train}}$ such that $|\mathcal{D}_\ell| = 0.01 \, |\mathcal{D}_{\text{train}}|$, corresponding to a 1% label regime. We compare the resulting performance across all SSL methods and a supervised model trained end-to-end using the same 1% labeled data. It is in this low-label regime that the benefits of self-supervised learning become most apparent. When averaged across all

datasets, all SSL methods, except InfoTS and SimMTM, outperform the fully supervised end-to-end model (Figure 1). Di-COT achieves the strongest overall performance, ranking first on three datasets and second-best on two others, highlighting its effectiveness in learning transferable and label-efficient representations (See Table 3).

Consistent with the results in Table 1, Random Forest (RF) attains the best performance on PAMAP2 even with limited labeled data. In contrast, MiniRocket, despite achieving strong results on WISDM2 when trained with full label supervision in the linear evaluation setting, performs worst in the low-label regime. This behavior indicates that MiniRocket's fixed feature representations are less robust when supervision is severely constrained. Overall, Di-COT attains the highest average accuracy of 76.36%, while being approximately $2.5\times$ faster to train than the second-best method, TF-C (73.55%). Both methods significantly outperform the supervised end-to-end model trained with 1% labels, which achieves an average accuracy of 70.39%.

### 4.3. Non-parametric $k$NN Evaluation

To assess the intrinsic quality of the learned representations independently of downstream optimization, we adopt a non-parametric $k$-nearest neighbor ($k$NN) evaluation protocol. This setting directly measures whether semantically similar samples are embedded in close proximity, without relying on classifier capacity or additional training. Given a pretrained encoder $f_\theta$, we extract embeddings for selected subsets of the training data as well as for the full test set. All embeddings are standardized, and classification is performed using euclidean distance in the representation space. We focus on the most local evaluation by setting $k = 1$, such that each test sample is assigned the label of its nearest training neighbor.

To study label efficiency, we vary the size of the labeled reference set and perform 1NN classification using subsets of $\{5, 10, 50, 100, 500\}$ randomly sampled labeled training

*Table 3.* Linear evaluation accuracy (%) across six large-scale datasets on 1% labeled data. Best in bold, second best underlined.

| | ECG | HARTH | PAMAP2 | SKODA | SLEEP | WISDM2 | Avg. Acc ± 95% CI |
|---|---|---|---|---|---|---|---|
| Supervised | 54.28 ± 3.22 | 75.37 ± 6.81 | 60.23 ± 1.82 | 92.77 ± 0.48 | 75.16 ± 0.29 | 64.56 ± 1.35 | 70.39 ± 1.00 |
| Raw Data | 51.67 ± 0.00 | 46.23 ± 0.00 | 57.17 ± 0.00 | 94.54 ± 0.00 | 35.58 ± 0.00 | 40.73 ± 0.00 | 54.32 ± 0.00 |
| Random Forest | 52.50 ± 0.00 | 84.59 ± 0.00 | **64.94 ± 0.00** | 92.58 ± 0.00 | 25.57 ± 0.00 | 41.98 ± 0.00 | 60.36 ± 0.00 |
| MiniRocket | 62.06 ± 7.10 | 19.86 ± 4.52 | 5.40 ± 1.21 | 17.87 ± 6.28 | 44.63 ± 0.54 | 40.38 ± 0.36 | 31.47 ± 2.75 |
| Random Init. | 56.56 ± 13.10 | 80.52 ± 7.28 | 59.42 ± 2.05 | 97.46 ± 0.43 | 74.18 ± 1.36 | 52.57 ± 2.80 | 70.12 ± 1.72 |
| TNC | 61.06 ± 7.08 | 83.04 ± 4.62 | 59.13 ± 2.79 | 96.11 ± 0.40 | 81.22 ± 0.88 | 55.83 ± 2.62 | 72.73 ± 1.68 |
| TS-TCC | 70.78 ± 6.46 | 81.25 ± 6.25 | 58.38 ± 1.52 | 91.24 ± 1.22 | 81.33 ± 0.85 | 55.88 ± 3.32 | 73.14 ± 1.92 |
| TS2Vec | 52.06 ± 7.76 | 86.49 ± 1.29 | 59.99 ± 1.66 | 96.29 ± 0.48 | 80.76 ± 0.80 | 56.31 ± 2.89 | 71.98 ± 1.28 |
| TF-C | **74.50 ± 5.94** | 78.00 ± 5.72 | 59.52 ± 2.75 | 93.50 ± 0.58 | **81.63 ± 0.77** | 54.13 ± 3.56 | 73.55 ± 1.34 |
| CoST | 68.17 ± 2.54 | 77.44 ± 7.19 | 59.89 ± 1.09 | 93.51 ± 0.58 | 80.18 ± 1.71 | **56.99 ± 5.30** | 72.70 ± 1.09 |
| InfoTS | 59.28 ± 10.51 | 72.00 ± 10.05 | 60.53 ± 2.86 | 97.25 ± 0.67 | 74.90 ± 1.36 | 53.56 ± 1.75 | 69.59 ± 1.68 |
| SimMTM | 47.33 ± 5.11 | 68.58 ± 5.65 | 58.66 ± 1.16 | 89.64 ± 0.78 | 81.33 ± 0.89 | 54.20 ± 3.54 | 66.62 ± 1.22 |
| Soft | 58.28 ± 11.20 | 85.78 ± 2.43 | 59.70 ± 1.99 | 95.23 ± 0.69 | 80.81 ± 0.85 | 55.51 ± 4.12 | 72.55 ± 1.43 |
| CaTT | 64.67 ± 5.14 | 85.09 ± 4.55 | 60.57 ± 3.56 | 83.20 ± 0.79 | 81.34 ± 0.71 | 54.84 ± 1.18 | 71.62 ± 1.07 |
| Di-COT | 73.33 ± 3.81 | **87.23 ± 1.95** | 61.90 ± 2.61 | **98.01 ± 0.16** | 81.38 ± 0.83 | 56.31 ± 1.86 | **76.36 ± 1.09** |

instances per class. For each subset size, embeddings of the selected samples constitute the reference set, while all test samples are classified based on their nearest neighbor in the embedding space. Results are averaged over five random seeds to mitigate sampling variance and are reported in Figure 4. As shown in Figure 4, Di-COT consistently outperforms competing methods across all label regimes under the $k$NN evaluation. Unlike linear probing, $k$NN classification is fully non-parametric and introduces no additional learnable parameters; strong performance therefore reflects the inherent organization of the learned representation rather than the expressiveness of a downstream classifier. The observed gains indicate that Di-COT produces embeddings with a well-structured local neighborhood, where semantically similar samples are mapped to close regions in the representation space and nearest neighbors are predominantly class-consistent (Full results in Appendix I).

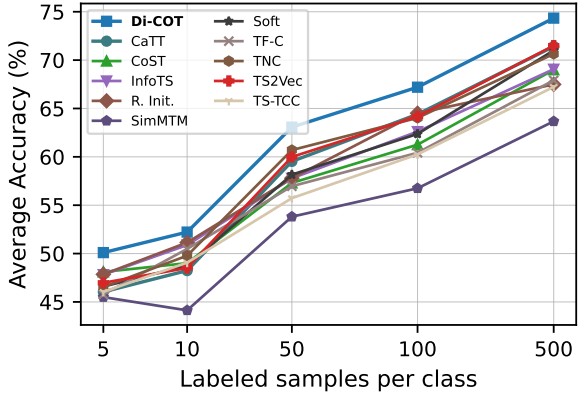

*Figure 4.* Non-parametric $k$NN evaluation of learned representations, averaged over five random seeds and six large-scale datasets.

### 4.4. Clustering

To complement linear probing and low-label evaluations, we further assess the global structure of the learned representations using clustering metrics. While linear probing

and $k$NN classification evaluate downstream discriminability and local neighborhood consistency, clustering measures whether semantically similar samples naturally form coherent groups in the embedding space without using labels during representation learning or clustering. We perform unsupervised $k$-means clustering on the embeddings produced by each method and evaluate cluster quality using Normalized Mutual Information (NMI) and Adjusted Rand Index (ARI). Although clustering itself is fully unsupervised, both metrics compare the resulting cluster assignments to ground-truth labels, thereby quantifying how well semantic class structure is preserved in the learned representation. Results are averaged over six datasets and five random seeds and summarized in Table 4.

Di-COT achieves the highest average NMI (0.508) and ARI (0.406), outperforming both self-supervised baselines and the fully supervised backbone. Its lower average rank (3.08) indicates that Di-COT consistently produces tighter and more semantically coherent clusters across datasets. These results are consistent with the $k$NN evaluation in Section 4.3: while 1NN highlights strong local neighborhood consistency, clustering demonstrates that this structure extends to global, class-level organization. Several baselines, such as TNC and CaTT, achieve moderate NMI but substantially lower ARI, suggesting that their embeddings capture partial class information but yield fragmented or overlapping clusters. In contrast, Di-COT consistently attains high scores on both metrics, indicating its ability to preserve semantic similarity while avoiding spurious splits. Notably, although CaTT ranks second in linear probing performance, it is ranked fifth in clustering, suggesting that some methods learn representations optimized for linear separability rather than globally well-structured embedding spaces. The clustering evaluation corroborates findings from linear probing and $k$NN experiments: Di-COT learns representations that are locally consistent, globally well-organized, and robust across datasets, making them well-suited for a wide range of downstream tasks (Full results in Appendix G).

Figure 5 in the Appendix presents t-SNE visualizations of the learned embeddings on the SKODA dataset. Di-COT exhibits more compact and class-consistent clusters with clearer inter-class separation compared to prior methods, qualitatively supporting the improvements observed in $k$NN accuracy and NMI/ARI scores.

*Table 4.* Clustering performance (NMI, ARI) averaged across five random seeds and six large-scale datasets.

| Metric | Di-COT | Sup_B | CoST | TNC | InfoTS | Soft | R. Init | CaTT | TF-C | TS2Vec | TS-TCC | SimMTM |
|---|---|---|---|---|---|---|---|---|---|---|---|---|
| NMI | **0.508** | 0.493 | 0.486 | 0.491 | 0.496 | 0.475 | 0.470 | 0.408 | 0.459 | 0.466 | 0.458 | 0.448 |
| ARI | **0.406** | 0.376 | 0.379 | 0.375 | 0.375 | 0.359 | 0.367 | 0.262 | 0.361 | 0.338 | 0.346 | 0.359 |
| AvgRank | **2.92** | 4.33 | 4.58 | 5.42 | 7.83 | 6.83 | 7.08 | 7.67 | 7.67 | 7.58 | 7.83 | 8.25 |

## 4.5. Cross-Dataset Transfer

To evaluate the generalization capability of the learned representations, we conduct cross-dataset transfer learning experiments. While linear probing and $k$NN evaluation assess performance within the same data distribution, and clustering measures intrinsic structure, cross-dataset transfer explicitly tests whether representations learned from one domain can generalize to unseen datasets with different sensor configurations and data distributions. For each source–target pair, we first pretrain the encoder in a self-supervised manner on the source dataset using only unlabeled data. The pretrained encoder is then frozen, and a linear classifier is trained on labeled samples from the target dataset. Performance is reported as accuracy on the target test set. This protocol isolates representation quality by preventing any adaptation of the encoder to the target domain. We evaluate cross-dataset transfer under four settings: HARTH $\rightarrow$ {PAMAP2, WISDM2} and WISDM2 $\rightarrow$ {PAMAP2, HARTH}.

A primary challenge in cross-dataset transfer learning stems from mismatched sensor dimensionality across datasets. While HARTH provides 6 channels, WISDM contains 3 channels, and PAMAP2 includes 52 channels, all three datasets share tri-axial accelerometer measurements. To establish semantic alignment, we restrict all datasets to a common three-channel accelerometer subspace corresponding to the $\{x, y, z\}$ axes. Specifically: WISDM is used as provided; HARTH's final three channels correspond to its tri-axial accelerometer; and channels 4–6 in PAMAP2 represent the accelerometer's $\{x, y, z\}$ axes. This alignment enables cross-dataset transfer over identical sensor modalities without requiring architectural modifications or dataset-specific preprocessing pipelines.

Table 5 summarizes cross-dataset transfer results across all source–target pairs. Di-COT consistently achieves the best performance in all transfer settings, resulting in the highest overall average accuracy. Despite marginal gains compared to in-domain pretraining, the improvements are consistent across all source–target pairs and exceed those achieved with randomly initialized backbones. These results indicate improved robustness to distribution shifts and stronger

domain generalization. This evaluation demonstrates that Di-COT learns representations that are less dependent on dataset-specific statistics and more reflective of underlying semantic activity patterns, making them well-suited for deployment in realistic cross-domain scenarios.

*Table 5.* Cross-dataset transfer accuracy (mean ± std over 5 seeds) on aligned 3-channel accelerometers. Di-COT consistently outperforms baselines, showing stronger domain generalization.

| | HARTH Pretraining | | WISDM2 Pretraining | | Avg. Acc. |
|---|---|---|---|---|---|
| | PAMAP2 | WISDM2 | HARTH | PAMAP2 | |
| Random Init. | 63.52 ± 1.29 | 59.97 ± 1.42 | 85.67 ± 3.94 | 63.52 ± 1.29 | 68.17 |
| CoST | 65.69 ± 1.31 | 60.08 ± 1.76 | 90.23 ± 0.74 | 66.61 ± 0.61 | 70.65 |
| InfoTS | 62.14 ± 0.97 | 60.06 ± 3.18 | 85.17 ± 2.98 | 62.97 ± 0.68 | 67.59 |
| SimMTM | 65.53 ± 0.88 | 62.64 ± 3.81 | 88.61 ± 1.48 | 65.97 ± 0.16 | 70.69 |
| Soft | 65.92 ± 0.75 | 60.22 ± 0.96 | 86.93 ± 3.17 | 66.33 ± 0.92 | 69.85 |
| TF-C | 65.49 ± 0.45 | 62.05 ± 2.97 | 89.81 ± 1.64 | 65.79 ± 0.53 | 70.79 |
| TNC | **66.30 ± 1.25** | 61.81 ± 1.42 | 90.13 ± 0.30 | 66.71 ± 0.64 | 71.24 |
| TS2Vec | 61.21 ± 1.32 | 57.50 ± 1.37 | 85.89 ± 5.41 | 66.88 ± 0.57 | 67.87 |
| TS-TCC | 65.80 ± 0.23 | 60.26 ± 1.50 | 86.77 ± 3.20 | 66.10 ± 0.44 | 69.73 |
| CaTT | 65.96 ± 0.47 | 62.12 ± 2.79 | 90.31 ± 0.91 | 65.88 ± 0.82 | 71.07 |
| Di-COT | **66.27 ± 0.82** | **63.45 ± 3.51** | **91.65 ± 1.05** | **67.11 ± 0.36** | **72.12** |

## 4.6. Visualizations of $t$SNE Embeddings on the SKODA dataset.

Figure 5 presents t-SNE projections of the learned representations on the SKODA test set. **Di-COT** consistently forms compact and well-separated clusters across activities, whereas all other methods, including the supervised backbone, exhibit noticeable cluster overlap or fragmentation. In particular, Di-COT is the only approach that cleanly separates the challenging red and gray classes, which remain partially entangled in competing representations. The clear margins and uniform compactness observed across clusters indicate that Di-COT learns a highly structured embedding space, capturing fine-grained temporal distinctions that are not fully recovered by existing self-supervised methods or standard supervised training.

## 4.7. Ablation Study

We conduct an extensive ablation study to evaluate the contribution of different components of Di-COT and justify our architectural choices, using large-scale (Large) datasets as well as the UCR and UEA benchmarks. First, we examine the effect of sub-block overlap. Including a 50% overlap consistently improves performance across all benchmarks, confirming that overlap improves both accuracy and stability, as shown in Table 6. Removing the temperature scaling has minor effects, slightly reducing Large ($-0.72\%$) and UEA ($-0.63\%$) while having negligible effect on the UCR. Consequently, we use a contrastive temperature of $\tau = 0.07$ for the large-scale datasets and the UEA benchmark in all experiments. For the UCR benchmark, we set $\tau = 1$ (i.e., no temperature scaling), as this yields comparable performance. Next, we analyze the window split strategy. Sampling sub-blocks uniformly per iteration outperforms using a fixed global split for large-scale and UEA datasets, as it

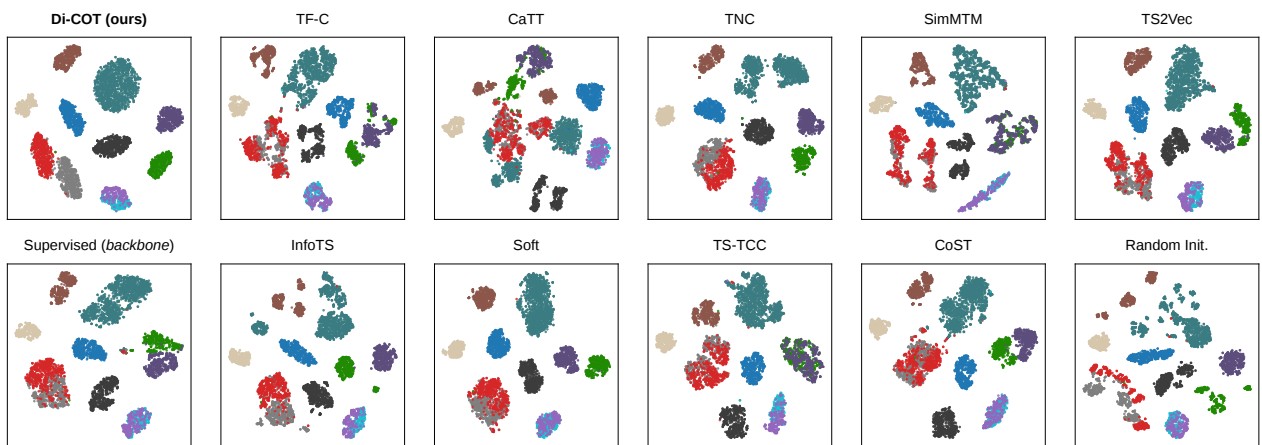

*Figure 5.* t-SNE visualizations of the learned embeddings on the SKODA test dataset across all self-supervised methods, supervised training, and random initialization.

allows the model to explore diverse temporal partitions. For UCR, a fixed global split performs reasonably well but at a higher computational cost (more details in Table 10 in the Appendix).

We also compare positive selection. Contrasting with the preceding sub-block consistently outperforms using shuffled sub-blocks, highlighting the importance of temporal context for learning meaningful representations. Replacing preceding sub-block contrast with next sub-block contrast results in only marginal performance differences across datasets (more details in Appendix B).

Replacing InceptionTime (Ismail Fawaz et al., 2020) with Residual Networks (ResNet) or Fully Convolutional Networks (FCN) in (Wang et al., 2017) consistently degrades performance across all datasets, with drops ranging from 2–6%, highlighting the impact of backbone choice. We additionally examine the use of a non-linear projection layer for contrastive learning. Unlike in computer vision (e.g., SimCLR), applying a non-linear projection does not improve performance for time series and can even slightly degrade results. The ablation study confirms that each design choice, sub-block overlap, preceding sub-block contrast, uniform per-iteration sampling, and backbone architecture, contributes positively to performance.

## 5. Conclusion

We present *Divide and Contrast* (Di-COT), an unsupervised framework for time-series representation learning that avoids data augmentations and multiple encoder passes by stochastically partitioning each time-series instance into overlapping sub-blocks and contrasting them in parallel. Di-COT reframes next-step prediction as a temporal contrastive classification task, maximizing translational robustness through dense, discriminative supervision in representation space rather than generative modeling in value space.

*Table 6.* Ablation results on large, UCR, and UEA datasets. Percentages indicate relative drop from the full Di-COT model.

| | Large | UCR | UEA |
|---|---|---|---|
| **Di-COT** | 83.07 | 81.33 | 71.24 |
| w/o Overlap | 81.22 (-2.23%) | 81.12 (-0.26%) | 70.13 (-1.56%) |
| w/o Temperature | 82.47 (-0.72%) | 81.33 (0.00%) | 70.79 (-0.63%) |
| *Window Split Strategy* | | | |
| Uniform Splits Sampling | | | |
| → Global Fixed Splits | 82.80 (-0.33%) | 81.32 (-0.01%) | 69.69 (-2.18%) |
| *Positive Selection* | | | |
| Preceding Sub-block Contrast | | | |
| → Next Sub-block Contrast | 82.22 (-1.02%) | 80.59 (-0.91%) | 70.70 (-0.76%) |
| → Shuffled Sub-block Contrast | 81.80 (-1.53%) | 81.19 (-0.17%) | 70.09 (-1.61%) |
| *Projection Layer* | | | |
| + Non-linear Projection | 81.85 (-1.47%) | 79.88 (-1.78%) | 69.73 (-2.12%) |
| *Backbone Architectures* | | | |
| InceptionTime | | | |
| → FCN | 80.93 (-2.58%) | 79.12 (-2.72%) | 69.50 (-2.44%) |
| → ResNet | 80.11 (-3.56%) | 76.76 (-5.62%) | 69.73 (-2.12%) |

Extensive experiments demonstrate that this design not only achieves state-of-the-art performance on large-scale real-world datasets, but also on the challenging UEA and UCR benchmarks, while requiring the lowest training time among self-supervised methods. Di-COT also excels in low-label regimes, outperforming both existing self-supervised approaches and end-to-end supervised models when only a small fraction of labeled data is available, highlighting its practical applicability in settings with scarce annotations. Furthermore, cross-dataset transfer experiments indicate that the learned representations generalize effectively across domains, suggesting potential for a single foundation model that can adapt in zero- or few-shot scenarios.

**Limitations.** Di-COT is designed around a contrastive objective that learns discriminative representations by drawing semantically similar instances closer in the embedding space, an inductive bias well-suited for classification, clustering, and retrieval tasks, but possibly orthogonal to forecasting tasks that require fine-grained temporal prediction. Extending Di-COT to such tasks may require a fundamentally different learning objective and is left for future work.

## Acknowledgments

This publication was funded by SFI NorwAI (Centre for Research-based Innovation, 309834) and the Office of Naval Research. SFI NorwAI is financially supported by its partners and the Research Council of Norway. The views expressed in this article are those of the author(s) and do not reflect the official policy or position of the U.S. Naval Academy, Department of the Navy, the Department of Defense, or the U.S. Government.

## Impact Statement

This paper presents work whose goal is to advance the field of Machine Learning. There are many potential societal consequences of our work, none which we feel must be specifically highlighted here.

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

**Code Availability** – To support reproducibility, the complete codebase and scripts for Di-COT are provided in the supplementary material and will be released publicly upon acceptance.

## A. Temporal Contrastive Loss

This appendix section provides a detailed perspective on Di-COT's objective by interpreting it as a form of classification induced by contrastive similarity scores. While the main paper presents Di-COT as a contrastive learning framework, this alternative view clarifies why the objective yields strong representations and why overlapping sub-blocks play a central role in the learning signal. The classification perspective reveals that Di-COT performs contrastive learning *within* a classification loss, bridging two successful paradigms in self-supervised learning.

### A.1. Formal Definition

Let $\mathcal{X} = \{\mathbf{x}_t\}_{t=1}^T$ be a multivariate time series of length $T$. Given an overlap parameter $\rho \in (0,1)$ and splits $k \sim \mathcal{U}\{2, 10\}$, we construct $k$ overlapping sub-blocks $\{\mathbf{W}_i\}_{i=0}^{k-1}$ with sub-block length $L$ where:

$$L = \frac{T}{1 + (k-1)(1-\rho)}. \tag{2}$$

To ensure symmetry in temporal pooling, we enforce $L$ to be even:

$$L \leftarrow 2 \cdot \left\lfloor \frac{L}{2} \right\rfloor. \tag{3}$$

The stride between consecutive sub-blocks is then defined as

$$s = \lfloor L(1-\rho) \rceil. \tag{4}$$

Since $k$ is specified as an expected number of sub-blocks, the actual number of extracted blocks may differ slightly depending on $T$. We therefore recompute the effective number of sub-blocks as

$$k \leftarrow \left\lfloor \frac{T-L}{s} \right\rfloor + 1. \tag{5}$$

Let $f_\theta$ be an encoder network mapping each sub-block to a $d$-dimensional embedding:

$$\mathbf{z}_i = f_\theta(\mathbf{W}_i) \in \mathbb{R}^d.$$

For an anchor sub-block index $j$, we compute similarity scores against all candidate sub-blocks in the same sequence:

$$s_{ji} = \mathrm{sim}(\mathbf{z}_j, \mathbf{z}_i) = \frac{\mathbf{z}_j^\top \mathbf{z}_i}{\tau}$$

where $\tau > 0$ is a temperature parameter controlling the hardness of the classification.

The Di-COT objective is then defined as:

$$\mathcal{L}_{\text{Di-COT}} = -\frac{1}{k} \sum_{j=0}^{k-1} \log \frac{\exp(s_{j,p^*(j)})}{\sum_{i=0}^{k-1} \exp(s_{ji})}$$

with the target mapping $p^* : \{0, \ldots, k-1\}$ defined as:

$$p^*(j) = \begin{cases} 0, & j = 0, \\ j-1, & j > 0. \end{cases}$$

## A.2. From Temporal Prediction to Similarity-Based Classification

Rather than predicting raw values or exact future sub-blocks, Di-COT encourages embeddings of temporally adjacent sub-blocks to be similar in representation space, while contrasting them against all other sub-blocks in the sequence. Specifically, given a sequence of overlapping sub-blocks $\{\mathbf{z}_j\}_{j=0}^{K-1}$, the task is to maximize the similarity between sub-block $\mathbf{z}_j$ and its preceding sub-block $\mathbf{z}_{j-1}$.

This formulation is enabled by the use of overlapping windows: the temporally adjacent sub-block already exists among the set of candidates, allowing temporal contrasting to be expressed as a classification problem over sub-block indices. In this sense, Di-COT replaces value-space prediction with representation-space discrimination, focusing on learning relational structure rather than generative modeling.

**Similarity Scores as Classification Logits.** For a given anchor $\mathbf{z}_j$, the similarity scores $\{s_{jk}\}_{k=0}^{K-1}$ are treated directly as logits in a $K$-way multi-class classification problem. The target class index $p^*(j)$ corresponds to the temporally adjacent (previous) sub-block. The softmax normalization over all $K$ candidates induces competition among them, with the objective encouraging the similarity of the correct adjacent sub-block to dominate.

The temperature parameter $\tau$ modulates this competition: smaller $\tau$ values sharpen the distribution, emphasizing difficult negatives and creating a harder classification task, while larger $\tau$ values soften the distribution, making the task easier by treating all candidates more equally. This provides a natural mechanism for controlling the difficulty of the pretext task.

**Contrastive Learning through Classification.** Although expressed as a classification loss, the objective remains inherently contrastive. To see this, consider the gradient with respect to a similarity score $s_{jk}$:

$$\frac{\partial \mathcal{L}}{\partial s_{jk}} = \begin{cases} p_{jk} - 1 < 0, & \text{if } k = p^*(j) \quad \text{(positive pair)}, \\ p_{jk} > 0, & \text{if } k \neq p^*(j) \quad \text{(negative pair)}, \end{cases}$$

where $p_{jk} = \exp(s_{jk})/\sum_{\ell=0}^{K-1} \exp(s_{j\ell})$ is the softmax probability.

The gradient structure reveals the contrastive nature:

- For the positive pair ($k = p^*(j)$), the gradient is negative, increasing the similarity $s_{j,p^*(j)}$ (pulling together).

- For negative pairs ($k \neq p^*(j)$), the gradient is positive, decreasing the similarities $s_{jk}$ (pushing apart).

Thus, the optimization structures the embedding space by pulling adjacent sub-blocks together while pushing all non-adjacent sub-blocks apart, exactly the effect of standard contrastive losses. This perspective highlights that Di-COT performs contrastive learning *within* the classification loss itself, making the distinction between contrastive and classification objectives largely notational.

## A.3. Overlapping Sub-Blocks

Di-COT contrasts adjacent sub-blocks and leverages overlapping sub-blocks for several complementary reasons that together create a well-posed and informative pretext task. Contrasting with the previous sub-block provides a consistent learning signal: all sub-blocks except the first have a clear preceding sub-block, while the first sub-block naturally references itself ($p^*(0) = 0$), providing a boundary condition.

Overlapping sub-blocks further enhance the learning signal by creating multiple slightly different training instances from a single sequence, effectively serving as implicit data augmentation. Each sub-block sees a unique temporal neighborhood, which encourages robustness to small temporal shifts, provides dense supervision (as every time step participates in multiple sub-blocks), and forces the encoder to learn features invariant to minor temporal translations. Together, these design choices yield a pretext task that is neither trivial nor excessively difficult, but encourages the model to learn discriminative representations capturing gradual temporal progression and state transitions.

## A.4. Relation to Existing Frameworks

Di-COT bridges two successful lines of work in self-supervised learning:

**Classification-based pretext tasks**: Methods like rotation prediction (Gidaris et al., 2018) and jigsaw puzzle solving (Noroozi & Favaro, 2016) use discrete classification tasks with handcrafted label spaces. Di-COT shares their simplicity and

ease of optimization but uses a *learned* similarity function rather than a fixed classifier, and its label space (sub-blocks) emerges naturally from the data structure and changes stochastically per iteration.

**Contrastive learning methods**: Approaches like SimCLR (Chen et al., 2020) and MoCo (He et al., 2020) use explicit positive/negative pair construction with the InfoNCE loss. Di-COT achieves similar embedding space structure but through temporal contrastive classification, bypassing the need for explicit pair mining and negative sampling strategies.

Di-COT is also related to temporal contrastive methods such as Temporal Neighborhood Coding (Tonekaboni et al., 2021), Contrast all The Time (Shamba et al., 2025) and Contrastive Predictive Coding (CPC; Oord et al., 2018), but differs in several key aspects. While the objective can be interpreted as InfoNCE, Di-COT adopts a temporal contrastive formulation over overlapping sub-blocks, where each sub-block serves as both a positive and a negative, and similarity scores are treated as logits. This contrasts with CPC, which employs an autoregressive model and an InfoNCE loss. Additionally, Di-COT performs stochastic partitioning and a cross-entropy reformulation specifically optimized for time-series efficiency, enabling dense supervision and faster training.

### A.5. Implications for Future Work

Viewing Di-COT through the lens of temporal contrastive classification reveals why it scales efficiently with sequence length and produces transferable representations. By framing temporal adjacency prediction as similarity-based classification, Di-COT captures the representation-structuring benefits of contrastive learning while optimizing a simple, stable cross-entropy objective.

The stochastic partitioning strategy, where the number of sub-blocks varies per iteration, exposes the model to multiple temporal granularities. This implicit multi-scale learning encourages the encoder to extract features that are robust across short and long temporal contexts, effectively capturing multi-scale dynamics without requiring an explicit hierarchical classification scheme.

Finally, while this work focuses on classification tasks, future research could explore how Di-COT representations transfer to forecasting, anomaly detection, and multimodal settings. More broadly, these insights highlight how careful pretext task design, temporal adjacency classification over overlapping windows, can yield strong, efficiently learned representations and may guide the development of future self-supervised methods across domains.

## B. Impact of Positive Pair Selection

Under our formulation (Eq.1), the cross-entropy objective treats each sub-block $j$ as an anchor with a single positive label $p^*(j) = j - 1$, and all other $k - 1$ sub-block positions as negatives. This is equivalent to a $k$-way classification problem over sub-block indices, which admits a clean and fully parallelizable matrix implementation via the similarity matrix $S^{(i)} \in \mathbb{R}^{k \times k}$.

Extending this to multiple positives per anchor, specifically, treating both $j - 1$ and $j + 1$ as positives simultaneously breaks the single-label cross-entropy formulation, since `F.cross_entropy` expects exactly one target index per anchor.

This leaves the alternative of computing multiple loss terms. Following your suggestion, we conducted additional ablation experiments to evaluate this design choice. The bidirectional approach has a slightly higher average score on some metrics, and slightly lower on others; overall, the results show no significant improvement when using bidirectional positives.

*Table 7.* Ablation study comparing different positive selection strategies.

| Positive Selection | UCR | UEA | Linear | Linear 1% | kNN (500) | NMI | ARI | Transfer |
|---|---|---|---|---|---|---|---|---|
| Preceding Sub-block | 81.33 | 71.24 | 83.07 | 76.36 | 74.34 | 0.508 | 0.406 | 72.12 |
| Next Sub-block | 80.59 | 70.70 | 82.22 | 75.94 | 74.18 | 0.505 | 0.403 | 72.49 |
| Bidirectional (Preceding + Next) | 81.10 | 70.38 | 82.66 | 76.42 | 74.30 | 0.509 | 0.408 | 71.98 |

Somewhat surprisingly, we also find that the additional computational overhead of computing two different loss terms using the next and preceding positives is not substantial. To be specific, on the large-scale benchmarks, the training time for bidirectional contrast is 10382 seconds, compared to 10374 seconds for preceding sub-block contrast.

Overall, these results demonstrate that the choice of constructing positive pairs from preceding sub-blocks, next sub-blocks,

or bidirectional does not meaningfully affect the quality of the learned representations across any evaluation setting.

## C. Algorithm and Computational Complexity

This section analyzes Di-COT's computational and memory complexity in comparison to temporally adjacent contrastive learning methods such as CaTT (Shamba et al., 2025).

### C.1. Temporal Adjacency-Based Contrastive Learning

Recent work has explored efficiency-driven contrastive learning for time series by exploiting temporal adjacency. In particular, CaTT contrasts representations at neighboring timesteps within a sliding window, treating temporally adjacent representations as positives and leveraging in-batch negatives. Formally, given a batch of $B$ sequences, each of length $T$, CaTT produces representations

$$\mathbf{Z} = \{\mathbf{z}_{b,t} \in \mathbb{R}^d \mid b \in [1, B], \ t \in [1, T]\},$$

and computes a multi-positive contrastive objective over all pairs of temporally adjacent timesteps.

This formulation requires constructing a similarity matrix

$$\mathbf{S} \in \mathbb{R}^{(BT) \times (BT)}, \quad S_{(b,t),(b',t')} = \frac{\langle \mathbf{z}_{b,t}, \mathbf{z}_{b',t'} \rangle}{\tau},$$

where positives correspond to temporally adjacent indices and all other entries across the entire batch act as negatives.

While this design eliminates explicit data augmentation and reduces encoder passes, it introduces two fundamental limitations: (i) a strong assumption that temporal adjacency at timestep resolution implies semantic similarity, and (ii) a quadratic dependency on both batch size and sequence length.

### C.2. Limitations of Timestep-Level Adjacency

The assumption that neighboring timesteps are semantically consistent is frequently violated in datasets with rapid event transitions or short temporal dependencies, such as those in the UCR and UEA benchmarks. In such settings, contrasting all consecutive timesteps encourages trivial alignment.

More critically, the timestep-level contrast induces a similarity matrix whose size scales as $(BT)^2$, leading to:

$$\text{Time Complexity: } \mathcal{O}(B^2 T^2 d), \qquad \text{Memory Complexity: } \mathcal{O}(B^2 T^2).$$

As a result, CaTT is highly sensitive to batch size. Increasing $B$ improves parallelism but quadratically increases both compute and memory costs, quickly becoming prohibitive for long windows.

Empirically, on the HARTH dataset, increasing the batch size from $B = 8$ to $B = 64$ results in a substantial degradation in training efficiency for CaTT: training becomes over $2\times$ slower than Di-COT, and larger batch sizes cannot be accommodated due to memory constraints (Figure 6). In contrast, Di-COT remains stable and efficient for batch sizes up to $B = 512$.

### C.3. Di-COT: sub-block-level Contrast

Di-COT addresses these limitations by relaxing strict timestep-level adjacency and operating on a set of $k \ll T$ sub-blocks extracted from each window. Given a sequence of length $T$, we partition it into $k$ contiguous sub-blocks

$$\mathcal{B} = \{\mathbf{x}^{(1)}, \dots, \mathbf{x}^{(k)}\},$$

and encode each sub-block into a representation $\mathbf{z}^{(i)} \in \mathbb{R}^d$.

Rather than contrasting all sub-blocks pairwise across the batch, Di-COT formulates a temporal contrastive objective in which each sub-block representation is trained to maximize similarity with a temporally adjacent target sub-block from the same instance (e.g., the preceding sub-block), while all remaining sub-blocks in the batch implicitly serve as negatives.

Importantly, the resulting similarity computation involves only a matrix of size

$$\mathbf{S}_{\text{Di-COT}} \in \mathbb{R}^{(Bk) \times k},$$

which can be efficiently implemented using batched tensor contractions (e.g., `einsum`) without materializing large dense matrices.

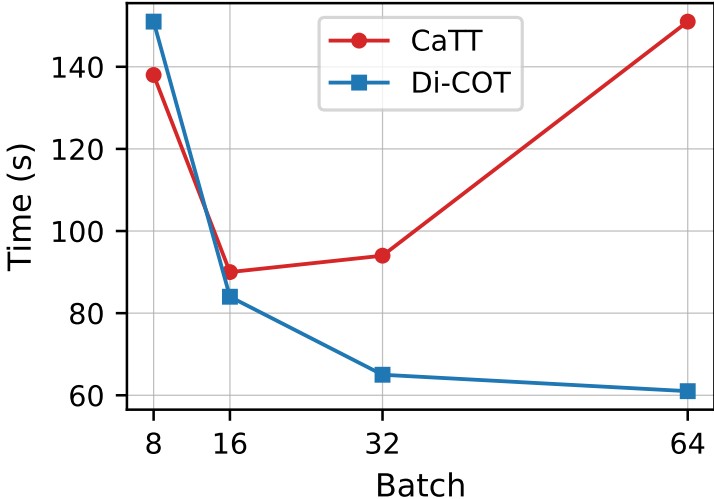

*Figure 6.* Comparison of efficiency between CaTT and Di-COT on increasing batch size.

### C.4. Computational Complexity Analysis

Under this formulation, Di-COT achieves:

$$\text{Time Complexity: } \mathcal{O}(Bk^2d), \qquad \text{Memory Complexity: } \mathcal{O}(Bk^2),$$

which is independent of the original sequence length $T$.

Since $k \ll T$ by design, this yields orders-of-magnitude savings in both computation and memory. Crucially, complexity scales linearly with batch size $B$, making Di-COT substantially more robust to large-batch training than timestep-level contrastive methods.

### C.5. Implications for Scalability and Generalization

By restricting contrastive learning to informative substructures rather than all timesteps, Di-COT resolves the efficiency–generalization trade-off inherent in temporally adjacent contrastive learning. sub-block-level contrast avoids learning trivial timestep-level information while enabling scalable training on long sequences and large batches.

This design choice explains both Di-COT's improved efficiency relative to CaTT and its superior performance on datasets where temporal continuity does not necessarily align with semantic consistency, as is the case for many datasets in the UCR and UEA archives.

## D. Reproducibility and Computational Environment

To ensure reproducibility, we report all experimental settings and computational details used in our study. Experiments on the large-scale benchmarks are conducted over five random seeds $\{1, 2, 3, 4, 5\}$, and results are reported as averages across seeds. For the UCR and UEA benchmarks, we follow prior work and use a fixed random seed of 1.

Unless stated otherwise, Di-COT uses a contrastive temperature of $\tau = 0.07$. Based on the ablation results in Table 10, temperature scaling has negligible impact on performance on UCR; therefore, we set $\tau = 1$ (i.e., no temperature scaling) for all UCR experiments.

The default sub-block overlap ratio is set to $50\%$. For uniform window sampling, we use $k_{\min} = 2$ and $k_{\max} = 10$ across all experiments. For ablation, the fixed global split is set to $k = 10$, as this value yields the strongest overall performance across datasets (Table 10).

All experiments are implemented in Python 3.12.3 using PyTorch 2.5.1 with CUDA 12.4. Experiments are executed on

NVIDIA Tesla V100S GPUs with 32 GB memory. The system uses NVIDIA driver version 570.172.08.

## E. Baseline Descriptions.

This appendix provides implementation details and reproduction settings for all baseline methods compared in this work. To ensure a controlled and fair comparison, we standardize the backbone architecture, data-loading pipeline, and optimization strategy across all methods. Specifically, all models are trained using a shared 1D InceptionTime backbone (Ismail Fawaz et al., 2020) and optimized with AdamW under a unified learning-rate schedule. Evaluating all methods under a fixed InceptionTime backbone (Ismail Fawaz et al., 2020) and shared training configuration ensures that performance differences reflect contrastive learning objectives rather than architectural choices. Unless otherwise stated, method-specific hyperparameters are taken from the official implementations released by the original authors, with no dataset-specific tuning performed.

All models are optimized using AdamW with a learning rate of $3 \times 10^{-4}$, weight decay of $3 \times 10^{-4}$, and momentum parameters $\beta_1 = 0.9$, $\beta_2 = 0.99$. We employ a cosine learning-rate schedule with linear warm-up during the first $10\%$ of training iterations. These optimization settings are applied uniformly across all methods.

**InfoTS** (Luo et al., 2023). We use the official implementation and default method-specific hyperparameters as described in the original work, including the combined global and local contrastive objectives.

**TS2Vec** (Yue et al., 2022). We adopt the official TS2Vec implementation with default settings.

**TNC** (Tonekaboni et al., 2021). We use the official implementation of TNC, including the original neighborhood sampling strategy and discriminator-based binary classification objective, with all method-specific hyperparameters set according to the authors' recommendations.

**CoST** (Woo et al., 2022). We reproduce CoST using the official implementation and default hyperparameters, including the original temporal-frequency decomposition and contrastive objective.

**SimMTM** (Dong et al., 2023). We use the official implementation of SimMTM with default masking and training configurations, without additional tuning.

**Soft** (Lee et al., 2024). Soft is not a standalone framework and is combined with TS2Vec following the authors' recommendation. We use the official implementation and adopt the cosine similarity option for computing soft assignments, as Dynamic Time Warping (DTW) is computationally prohibitive for long sequences.

**TS-TCC** (Eldele et al., 2021). We use the official implementation of TS-TCC with default hyperparameters, following the original temporal and contextual contrastive learning formulation.

**TF-C** (Zhang et al., 2022). We adopt the official TF-C implementation with default settings, jointly contrasting time-domain and frequency-domain representations.

**CaTT** (Shamba et al., 2025). We use the official implementation of CaTT with the default training protocol, which contrasts temporally adjacent and repeated time steps without explicit data augmentations.

**MiniRocket** (Dempster et al., 2021). MiniRocket is included as a strong non-neural baseline for time-series classification. We use the official `sktime` implementation, transforming each time series into a fixed 10,000-dimensional feature representation for downstream evaluation.

## F. Detailed Dataset Information.

We use publicly available time series datasets previously employed in self-supervised representation learning research. The WISDM2, PAMAP2, SLEEP and SKODA datasets were directly obtained from the repository linked by the Series2Vec project, without applying additional preprocessing. For the ECG dataset, we adopted a TNC-style preprocessing to generate random windows for self-supervised learning. Each subject's continuous 12-hour recording was split into 10-minute segments to reduce redundancy, and during training, a random window of 2,500 time steps was sampled from each segment. The first 5 subjects were held out for testing, with the remaining subjects used for training. This strategy increases variability in the training data while maintaining consistent input dimensions across subjects. For linear probing, we extract a deterministic window of 2,500 time steps from the middle of each segment to ensure reproducible evaluation. For kNN

evaluation, we instead use the full subject-wise split without segmenting into 10-minute windows, allowing access to all available data for the limited labeled samples (5, 10, 50, 100, 500) while preserving the original distribution. For the HARTH dataset, we performed subject-wise split with 22 subjects in total. The first 18 subjects were used for training and the remaining 4 for testing. Each subject's multivariate time series was split into overlapping windows of 500 time steps with a stride of 250, and the majority class within each window was assigned as the label.

Table 8 summarizes the key characteristics of each dataset. All datasets are split using subject-wise splitting, the SKODA dataset, collected from a single subject, however, does not have subject-wise splits.

*Table 8.* Summary of datasets used in the large-scale experiments.

| Dataset | Train set | Test set | Subject | Sequence Length | Channel | Class |
|---|---|---|---|---|---|---|
| ECG | 59,922 | 16,645 | 23 | 2,500 | 2 | 4 |
| HARTH | 23,025 | 2,785 | 22 | 500 | 6 | 18 |
| PAMAP2 | 51,192 | 11,590 | 9 | 100 | 52 | 18 |
| SKODA | 22,587 | 5,646 | 1 | 50 | 64 | 11 |
| SLEEP | 25,612 | 8,910 | 20 | 3,000 | 1 | 5 |
| WISDM2 | 134,614 | 14,421 | 29 | 40 | 3 | 6 |

## G. Full Clustering Results.

*Table 9.* Per-dataset clustering performance (NMI, ARI) with averages across all 6 datasets. Best in bold, second best underlined.

| Method | ECG | | HARTH | | SKODA | | WISDM2 | | PAMAP2 | | SLEEP | | Average | |
|---|---|---|---|---|---|---|---|---|---|---|---|---|---|---|
| | NMI | ARI | NMI | ARI | NMI | ARI | NMI | ARI | NMI | ARI | NMI | ARI | NMI | ARI |
| Random Init. | 0.263 | 0.285 | 0.675 | 0.466 | 0.910 | 0.825 | 0.144 | 0.104 | 0.575 | 0.378 | 0.251 | 0.143 | 0.470 | 0.367 |
| Supervised_B | 0.291 | 0.321 | 0.665 | 0.423 | 0.866 | 0.774 | 0.227 | 0.184 | 0.600 | 0.377 | **0.308** | **0.178** | 0.493 | 0.376 |
| CoST | 0.261 | 0.308 | 0.684 | 0.491 | 0.843 | 0.706 | 0.252 | 0.210 | 0.591 | **0.395** | 0.281 | 0.163 | 0.486 | 0.379 |
| InfoTS | 0.248 | 0.271 | 0.653 | 0.441 | 0.914 | 0.847 | 0.133 | 0.089 | 0.580 | 0.374 | 0.252 | 0.146 | 0.496 | 0.375 |
| SimMTM | 0.296 | **0.448** | 0.613 | 0.376 | 0.824 | 0.719 | 0.153 | 0.106 | 0.540 | 0.347 | 0.264 | 0.156 | 0.448 | 0.359 |
| Soft | 0.237 | 0.232 | 0.696 | 0.505 | 0.870 | 0.746 | 0.212 | 0.159 | 0.577 | 0.360 | 0.255 | 0.154 | 0.475 | 0.359 |
| TF-C | 0.224 | 0.267 | 0.652 | 0.432 | 0.818 | 0.760 | 0.232 | 0.216 | 0.555 | 0.337 | 0.271 | 0.157 | 0.459 | 0.361 |
| TNC | 0.253 | 0.336 | 0.662 | 0.407 | 0.877 | 0.745 | **0.294** | **0.249** | 0.567 | 0.346 | 0.295 | 0.169 | 0.491 | 0.375 |
| TS2Vec | 0.201 | 0.171 | 0.677 | 0.466 | 0.861 | 0.722 | 0.210 | 0.148 | 0.582 | 0.365 | 0.263 | 0.156 | 0.466 | 0.338 |
| TS-TCC | 0.276 | 0.296 | 0.674 | 0.451 | 0.818 | 0.713 | 0.170 | 0.131 | 0.530 | 0.318 | 0.279 | 0.167 | 0.458 | 0.346 |
| CaTT | **0.309** | 0.389 | 0.646 | 0.393 | 0.455 | 0.183 | 0.238 | 0.187 | 0.533 | 0.258 | 0.268 | 0.163 | 0.408 | 0.262 |
| Di-COT | 0.292 | 0.290 | **0.717** | **0.516** | **0.929** | **0.904** | 0.193 | 0.160 | **0.619** | 0.394 | 0.297 | 0.171 | **0.508** | **0.406** |

## H. Extended Ablation and Hyperparameter Analysis.

This appendix details the hyperparameter selection strategy for our method and provides additional ablation results. Our goal is to justify the default configuration used throughout the paper by highlighting the trade-offs between representation quality, computational efficiency, and scalability with respect to sequence length.

**Ablation protocol.** For each hyperparameter, we fix all remaining components to their default values and vary only the parameter under consideration. This controlled setup isolates the impact of each design choice.

**Default configuration.** Unless stated otherwise, we use a temperature of $\tau = 0.07$ for the UEA and Large datasets, and $\tau = 1.0$ for UCR (i.e., no temperature scaling). We fix the overlap ratio between adjacent sub-blocks to 50% and adopt uniform split sampling with $k_{max} = 10$. For all uniform sampling experiments, we fix $k_{min} = 2$, as at least two sub-blocks are required to define a meaningful in-sequence contrastive objective.

**Temperature scaling ($\tau$).** To analyze the effect of temperature scaling, we vary $\tau \in \{0.07, 0.1, 0.5, 1.0\}$ while fixing the overlap to 50% and using uniform split sampling with $k_{max} = 10$. As shown in Table 6, smaller temperatures generally

yield better performance on UEA and Large, while UCR is relatively insensitive to temperature scaling. This supports our choice of dataset-dependent defaults and shows that competitive performance can be achieved without careful tuning, reinforcing the robustness of the method.

**Overlap ratio.** We next study the impact of subblock overlap by varying the overlap ratio in $\{0.00, 0.25, 0.50, 0.75\}$, while fixing the temperature ($\tau = 0.07$ for UEA and Large, $\tau = 1.0$ for UCR) and using uniform split sampling with $k_{\max} = 10$. Moderate overlap (50%) consistently provides the best performance, suggesting that partial temporal sharing between sub-blocks improves representation learning without introducing excessive redundancy.

**Number of splits and computational efficiency.** We compare two subseries generation strategies: (1) *Global Fixed Splits*, where each series is split into exactly $k$ equal-length segments, and (2) *Uniform Split Sampling*, where $k$ is randomly sampled between $k_{\min} = 2$ and $k_{\max}$ per iteration. As shown in Table 10, increasing the number of splits generally improves performance, but also increases pretraining time due to the larger number of contrasted sub-blocks (Figure 7). Uniform sampling with $k_{\max} = 10$ achieves the best overall performance across datasets while remaining more efficient than larger values such as $k_{\max} = 15$ or 20, indicating that overly fine partitioning yields diminishing returns. Moreover, uniform splits per iteration with $k_{\max} = 10$ consistently matches or outperforms fixed $k$ values while requiring lesser training time. This demonstrates that introducing variability in the number of splits during pretraining can improve generalization while keeping training efficient.

**Implicit multi-scale learning**: The stochastic partitioning strategy, where the number of sub-blocks varies per iteration, exposes the model to multiple temporal granularities. This encourages the encoder to learn features that are robust across short and long temporal contexts, effectively capturing multi-scale dynamics without requiring an explicit hierarchical classification scheme.

*Table 10.* Additional ablation results and hyperparameter analysis on the Large, UCR, and UEA datasets.

|  | **Large** | **UCR** | **UEA** |
|---|---|---|---|
| Temperature $\tau$ | | | |
| $\rightarrow 0.07$ | 83.07 | 81.00 | 71.24 |
| $\rightarrow 0.1$ | 82.98 | 80.93 | 71.09 |
| $\rightarrow 0.5$ | 82.72 | 81.28 | 70.67 |
| $\rightarrow 1.0$ | 82.47 | 81.33 | 70.79 |
| Overlap | | | |
| $\rightarrow 0\%$ | 81.22 | 81.12 | 70.13 |
| $\rightarrow 25\%$ | 82.15 | 81.25 | 70.98 |
| $\rightarrow 50\%$ | 83.07 | 81.33 | 71.24 |
| $\rightarrow 75\%$ | 82.52 | 80.99 | 70.74 |
| Global Fixed Splits ($k$) | | | |
| $\rightarrow 2$ | 80.93 | 80.68 | 69.88 |
| $\rightarrow 5$ | 81.29 | 80.63 | 70.80 |
| $\rightarrow 10$ | 82.80 | 81.32 | 69.69 |
| $\rightarrow 15$ | 81.48 | 80.89 | 70.15 |
| Uniform Splits Sampling ($k_{max}$) | | | |
| $\rightarrow 5$ | 81.91 | 80.94 | 69.83 |
| $\rightarrow 10$ | 83.07 | 81.33 | 71.24 |
| $\rightarrow 15$ | 82.48 | 80.84 | 69.85 |
| $\rightarrow 20$ | 82.03 | 80.75 | 70.06 |

# I. Non-parametric $k$NN Evaluation Full Results.

In the main paper, we present a line chart summarizing the average classification accuracy across datasets as the number of labeled samples per class varies. For completeness and transparency, we provide here the full numerical results for each dataset at 5, 10, 50, 100, and 500 labeled samples per class. Each table reports the per-dataset accuracy for all compared methods, with the best results highlighted in **bold**.

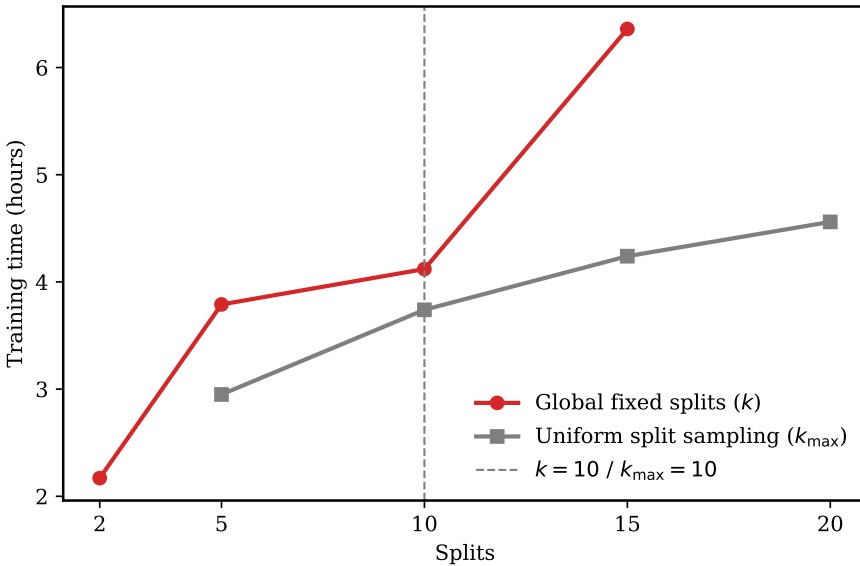

*Figure 7.* **Cumulative training time versus number of sub-block splits.** The figure reports the total pretraining time aggregated across the Large, UCR, and UEA datasets for global fixed splits ($k$) and uniform split sampling ($k_{max}$). Training time increases with finer partitioning due to the larger number of contrasted subblocks. The dashed line indicates the default setting ($k_{max} = 10$), which we adopt as a practical balance between computational cost and empirical performance.

*Table 11.* Per-dataset accuracy with 5 labeled samples per class.

| | ECG | HARTH | PAMAP2 | SKODA | SLEEP | WISDM2 | Avg Acc |
|---|---|---|---|---|---|---|---|
| Random Init. | $31.00 \pm 2.71$ | $47.62 \pm 8.08$ | $41.50 \pm 0.81$ | $91.57 \pm 0.88$ | $48.42 \pm 3.17$ | $26.96 \pm 1.35$ | 47.85 |
| CaTT | $24.61 \pm 0.93$ | $43.30 \pm 1.94$ | $39.94 \pm 0.66$ | $85.18 \pm 1.64$ | $54.40 \pm 0.68$ | $28.82 \pm 2.46$ | 46.04 |
| CoST | $30.72 \pm 2.30$ | $47.82 \pm 8.27$ | $40.36 \pm 2.09$ | $85.20 \pm 1.56$ | $53.55 \pm 1.54$ | $30.95 \pm 2.38$ | 48.10 |
| InfoTS | $30.67 \pm 3.21$ | $45.08 \pm 7.53$ | $41.54 \pm 2.82$ | $91.85 \pm 0.71$ | $52.30 \pm 1.86$ | $25.47 \pm 0.93$ | 47.82 |
| SimMTM | $30.56 \pm 2.89$ | $42.55 \pm 3.54$ | $38.27 \pm 1.17$ | $81.26 \pm 0.88$ | $54.18 \pm 0.36$ | $26.17 \pm 2.39$ | 45.50 |
| Soft | $28.39 \pm 1.67$ | $40.27 \pm 2.51$ | $40.29 \pm 1.44$ | $88.21 \pm 1.49$ | $53.97 \pm 0.54$ | $28.41 \pm 0.82$ | 46.59 |
| TF-C | $27.61 \pm 3.70$ | $43.53 \pm 4.58$ | $40.31 \pm 1.18$ | $82.09 \pm 2.49$ | $54.49 \pm 0.73$ | $26.61 \pm 1.27$ | 45.77 |
| TNC | $21.72 \pm 1.49$ | $44.89 \pm 2.34$ | $40.39 \pm 1.61$ | $89.61 \pm 0.86$ | $54.47 \pm 0.62$ | $28.23 \pm 2.22$ | 46.55 |
| TS2Vec | $30.56 \pm 4.96$ | $39.94 \pm 4.15$ | $41.00 \pm 0.90$ | $88.05 \pm 0.99$ | $54.18 \pm 0.77$ | $28.14 \pm 0.93$ | 46.98 |
| TS-TCC | $26.89 \pm 1.10$ | $40.65 \pm 5.48$ | $40.07 \pm 1.48$ | $81.39 \pm 1.83$ | $54.44 \pm 0.84$ | $27.77 \pm 2.10$ | 45.20 |
| Di-COT | $\mathbf{31.00 \pm 1.35}$ | $\mathbf{55.35 \pm 7.04}$ | $\mathbf{44.20 \pm 0.61}$ | $\mathbf{92.85 \pm 0.44}$ | $\mathbf{54.49 \pm 0.67}$ | $\mathbf{30.95 \pm 2.26}$ | $\mathbf{50.10}$ |

*Table 12.* Per-dataset accuracy with 10 labeled samples per class.

| | ECG | HARTH | PAMAP2 | SKODA | SLEEP | WISDM2 | Avg Acc |
|---|---|---|---|---|---|---|---|
| Random Init. | $45.56 \pm 6.53$ | $54.84 \pm 5.95$ | $41.14 \pm 1.82$ | $92.47 \pm 0.43$ | $44.29 \pm 0.97$ | $28.57 \pm 2.29$ | 51.14 |
| CaTT | $35.06 \pm 2.30$ | $53.24 \pm 1.55$ | $41.90 \pm 1.20$ | $87.41 \pm 1.13$ | $48.20 \pm 1.43$ | $23.63 \pm 1.02$ | 48.24 |
| CoST | $37.72 \pm 9.96$ | $50.45 \pm 5.88$ | $40.81 \pm 0.58$ | $87.16 \pm 0.96$ | $48.35 \pm 1.00$ | $29.57 \pm 1.55$ | 49.01 |
| InfoTS | $36.94 \pm 9.62$ | $57.45 \pm 6.72$ | $43.00 \pm 2.97$ | $92.82 \pm 0.47$ | $46.26 \pm 2.19$ | $28.82 \pm 1.86$ | 50.88 |
| SimMTM | $36.06 \pm 1.64$ | $37.95 \pm 8.14$ | $41.06 \pm 0.55$ | $82.40 \pm 0.19$ | $46.34 \pm 1.41$ | $20.96 \pm 3.00$ | 44.13 |
| Soft | $42.39 \pm 7.34$ | $47.61 \pm 1.31$ | $41.56 \pm 1.36$ | $90.12 \pm 1.23$ | $45.54 \pm 1.04$ | $25.25 \pm 2.01$ | 48.75 |
| TF-C | $48.28 \pm 6.29$ | $53.20 \pm 8.40$ | $40.27 \pm 2.61$ | $84.90 \pm 1.90$ | $48.25 \pm 1.36$ | $27.97 \pm 2.34$ | 50.48 |
| TNC | $40.78 \pm 4.16$ | $51.62 \pm 3.10$ | $41.94 \pm 1.25$ | $90.86 \pm 1.09$ | $48.04 \pm 1.61$ | $25.47 \pm 1.52$ | 49.78 |
| TS2Vec | $40.72 \pm 11.35$ | $47.43 \pm 0.94$ | $42.36 \pm 0.82$ | $91.03 \pm 1.04$ | $45.62 \pm 1.17$ | $23.87 \pm 1.95$ | 48.51 |
| TS-TCC | $48.83 \pm 3.69$ | $49.73 \pm 1.46$ | $40.98 \pm 2.73$ | $81.84 \pm 1.71$ | $48.46 \pm 1.35$ | $24.54 \pm 2.13$ | 49.06 |
| Di-COT | $\mathbf{48.67 \pm 2.40}$ | $\mathbf{63.66 \pm 6.91}$ | $\mathbf{43.11 \pm 1.21}$ | $\mathbf{95.13 \pm 0.51}$ | $\mathbf{48.49 \pm 1.41}$ | $\mathbf{30.57 \pm 1.19}$ | $\mathbf{52.22}$ |

*Table 13.* Per-dataset accuracy with 50 labeled samples per class.

|  | ECG | HARTH | PAMAP2 | SKODA | SLEEP | WISDM2 | Avg Acc |
|---|---|---|---|---|---|---|---|
| Random Init. | 42.56 ± 4.84 | 61.45 ± 5.59 | 59.11 ± 1.07 | 96.29 ± 0.43 | 48.54 ± 1.44 | 38.78 ± 1.21 | 57.79 |
| CaTT | 43.61 ± 3.86 | 66.28 ± 1.08 | 59.84 ± 0.43 | 90.39 ± 1.57 | 57.35 ± 1.29 | 39.72 ± 2.09 | 59.53 |
| CoST | 30.44 ± 2.56 | 64.04 ± 1.90 | 58.38 ± 0.21 | 90.32 ± 1.04 | 57.98 ± 0.84 | 42.67 ± 2.30 | 57.30 |
| InfoTS | 39.72 ± 2.78 | 61.39 ± 5.42 | 59.26 ± 1.17 | 96.50 ± 0.32 | 50.15 ± 2.08 | 39.67 ± 2.56 | 57.78 |
| SimMTM | 37.61 ± 1.09 | 51.65 ± 6.18 | 56.17 ± 0.86 | 85.59 ± 0.78 | 55.81 ± 1.02 | 36.00 ± 3.75 | 53.81 |
| Soft | 35.72 ± 3.61 | 64.79 ± 1.14 | 59.17 ± 0.53 | 92.14 ± 1.24 | 55.61 ± 1.99 | 41.50 ± 1.99 | 58.16 |
| TF-C | 39.22 ± 5.35 | 54.51 ± 6.29 | 59.34 ± 1.44 | 90.09 ± 0.94 | 57.26 ± 1.25 | 41.37 ± 2.75 | 56.96 |
| TNC | 42.11 ± 4.00 | 70.01 ± 1.39 | 59.13 ± 0.81 | 93.16 ± 0.83 | 57.60 ± 1.38 | 42.19 ± 1.73 | 60.70 |
| TS2Vec | 44.33 ± 10.40 | 65.32 ± 1.49 | 59.42 ± 0.78 | 93.65 ± 0.71 | 55.67 ± 0.96 | 41.52 ± 2.77 | 59.99 |
| TS-TCC | 37.78 ± 1.32 | 53.78 ± 2.64 | 56.95 ± 0.71 | 86.99 ± 1.04 | 57.39 ± 1.12 | 41.47 ± 2.67 | 55.73 |
| Di-COT | **48.67 ± 4.05** | **81.93 ± 2.72** | **60.41 ± 0.73** | **97.06 ± 0.29** | **57.36 ± 1.29** | **42.06 ± 2.57** | **63.07** |

*Table 14.* Per-dataset accuracy with 100 labeled samples per class.

|  | ECG | HARTH | PAMAP2 | SKODA | SLEEP | WISDM2 | Avg Acc |
|---|---|---|---|---|---|---|---|
| Random Init. | 53.11 ± 2.79 | 74.80 ± 4.27 | 65.30 ± 0.66 | 96.35 ± 0.40 | 50.54 ± 1.65 | 46.96 ± 1.05 | 64.51 |
| CaTT | 53.72 ± 3.69 | 72.22 ± 2.37 | 65.83 ± 0.25 | 90.67 ± 1.30 | 57.23 ± 1.02 | 46.75 ± 2.20 | 64.40 |
| CoST | 36.72 ± 5.05 | 69.69 ± 2.37 | 65.69 ± 2.91 | 91.04 ± 0.62 | 58.96 ± 1.13 | 45.47 ± 1.91 | 61.26 |
| InfoTS | 45.56 ± 6.93 | 70.63 ± 3.22 | 65.44 ± 0.76 | 96.50 ± 0.28 | 51.43 ± 1.81 | 46.12 ± 1.42 | 62.61 |
| SimMTM | 37.28 ± 4.06 | 51.94 ± 2.52 | 61.82 ± 0.71 | 86.99 ± 0.94 | 56.80 ± 0.57 | 45.64 ± 1.21 | 56.75 |
| Soft | 43.39 ± 4.34 | 69.91 ± 4.80 | 65.17 ± 0.36 | 92.38 ± 0.63 | 57.43 ± 0.95 | 45.93 ± 1.62 | 62.37 |
| TF-C | 43.11 ± 4.49 | 60.83 ± 2.10 | 64.95 ± 0.68 | 90.65 ± 0.79 | 57.15 ± 1.11 | 45.81 ± 2.48 | 60.42 |
| TNC | 46.39 ± 8.88 | 75.64 ± 3.99 | 65.47 ± 0.62 | 93.49 ± 0.70 | 57.30 ± 1.08 | 45.91 ± 0.48 | 64.03 |
| TS2Vec | 50.94 ± 3.08 | 70.50 ± 5.90 | 65.63 ± 0.23 | 94.33 ± 0.55 | 57.48 ± 0.95 | 46.15 ± 1.33 | 64.17 |
| TS-TCC | 43.89 ± 1.13 | 61.00 ± 5.74 | 63.24 ± 0.54 | 87.80 ± 1.24 | 57.21 ± 0.89 | 48.38 ± 2.07 | 60.25 |
| Di-COT | **49.39 ± 2.80** | **84.30 ± 1.67** | **66.07 ± 0.29** | **97.27 ± 0.17** | **57.22 ± 1.02** | **48.93 ± 2.43** | **67.20** |

*Table 15.* Per-dataset accuracy with 500 labeled samples per class.

|  | ECG | HARTH | PAMAP2 | SKODA | SLEEP | WISDM2 | Avg Acc |
|---|---|---|---|---|---|---|---|
| Random Init. | 57.22 ± 7.48 | 74.89 ± 5.31 | 70.86 ± 0.60 | 97.26 ± 0.27 | 55.72 ± 0.89 | 48.99 ± 1.40 | 67.49 |
| CaTT | 72.44 ± 1.62 | 77.03 ± 2.41 | 71.67 ± 0.30 | 93.10 ± 1.29 | 65.14 ± 1.78 | 49.06 ± 1.67 | 71.41 |
| CoST | 59.00 ± 4.29 | 77.82 ± 4.77 | 70.19 ± 0.71 | 92.42 ± 0.78 | 65.02 ± 0.62 | 49.55 ± 1.51 | 69.00 |
| InfoTS | 63.28 ± 1.77 | 77.79 ± 6.03 | 72.00 ± 2.56 | 97.47 ± 0.13 | 56.19 ± 2.19 | 47.59 ± 0.92 | 69.06 |
| SimMTM | 58.61 ± 3.85 | 54.24 ± 5.05 | 69.31 ± 1.15 | 88.81 ± 0.67 | 63.85 ± 1.32 | 47.23 ± 0.71 | 63.67 |
| Soft | 66.33 ± 1.64 | 80.41 ± 5.44 | 71.68 ± 1.83 | 93.95 ± 0.80 | 63.40 ± 1.13 | 50.24 ± 0.88 | 71.00 |
| TF-C | 64.11 ± 0.63 | 64.22 ± 1.39 | 71.66 ± 1.87 | 92.48 ± 0.62 | 65.03 ± 1.66 | 49.97 ± 1.41 | 67.91 |
| TNC | 60.72 ± 4.21 | 80.81 ± 4.59 | 71.22 ± 0.88 | 94.78 ± 0.73 | 65.35 ± 1.76 | 50.99 ± 1.62 | 70.64 |
| TS2Vec | 66.67 ± 1.64 | 81.75 ± 3.39 | 71.41 ± 1.43 | 95.60 ± 0.84 | 63.56 ± 1.19 | 50.00 ± 1.23 | 71.50 |
| TS-TCC | 60.61 ± 6.51 | 67.77 ± 5.64 | 70.85 ± 1.77 | 89.40 ± 0.97 | 65.12 ± 1.66 | 49.60 ± 1.67 | 67.23 |
| Di-COT | **70.89 ± 1.07** | **89.63 ± 1.76** | **72.46 ± 1.30** | **97.94 ± 0.19** | **65.17 ± 1.78** | **49.95 ± 1.63** | **74.34** |

# J. Extended UCR & UEA Benchmarks Results.

The UEA archive originally contains 30 datasets. In our experiments, we report results on 28 datasets, excluding *InsectWingbeat* and *EigenWorms*. These two datasets exhibit atypical input shapes and extremely large memory requirements, which lead to runtime or memory errors for several baseline methods despite consistent preprocessing and hyperparameter settings. The UCR archive contains 128 datasets. We evaluate on 124 datasets, excluding *DodgerLoopDay*, *DodgerLoopGame*, *DodgerLoopWeekend*, and *MelbournePedestrian*, which consistently produce NaN values or runtime errors across multiple baselines despite standard normalization and training procedures.

We note that this behavior is not unique to our evaluation protocol. Prior work such as TS2Vec (Yue et al., 2022) similarly omits a comparable subset of datasets, reporting results on 125 UCR and 29 UEA datasets, due to analogous issues related to input dimensionality, memory constraints, or training instability.

*Table 16.* 28 UEA - Performance across datasets. Best per dataset in bold, second best underlined.

| dataset | Di-COT | CoST | SimMTM | Soft | TF-C | TNC | TS2Vec | TS-TCC | CaTT | MiniRocket | Random Forest | InfoTS |
|---|---|---|---|---|---|---|---|---|---|---|---|---|
| ArticularyWordRecognition | 95.67 | **97.67** | 94.33 | 97.00 | 97.33 | 97.33 | 97.33 | 97.33 | 95.00 | 6.67 | 3.67 | 95.67 |
| AtrialFibrillation | **40.00** | 33.33 | 13.33 | 20.00 | 33.33 | 26.67 | 26.67 | 26.67 | 26.67 | 26.67 | 33.33 | 33.33 |
| BasicMotions | 100.00 | 100.00 | 100.00 | 100.00 | 100.00 | 100.00 | 100.00 | 100.00 | 92.50 | 65.00 | 100.00 | 100.00 |
| CharacterTrajectories | 99.30 | 99.37 | 98.68 | **99.51** | **99.51** | 99.37 | 99.30 | 99.30 | 99.23 | 30.99 | 54.94 | 97.91 |
| Cricket | 98.61 | 98.61 | 90.28 | 97.22 | 95.83 | **100.00** | 97.22 | 97.22 | 95.83 | 37.50 | 9.72 | 98.61 |
| DuckDuckGeese | 56.00 | 56.00 | 60.00 | 54.00 | **62.00** | 54.00 | 52.00 | 52.00 | 50.00 | 26.00 | 48.00 | 58.00 |
| ERing | 89.63 | 87.04 | 90.37 | **91.85** | 90.37 | 88.52 | 90.00 | 85.56 | 89.63 | 37.04 | 15.93 | 80.74 |
| Epilepsy | **95.65** | 94.20 | 92.03 | 94.20 | 92.03 | **95.65** | 93.48 | 92.75 | 89.13 | 73.91 | 89.86 | 93.48 |
| EthanolConcentration | 23.95 | **41.06** | 28.52 | 29.66 | 26.24 | 25.86 | 26.24 | 26.24 | 28.90 | 25.10 | 29.66 | 30.80 |
| FaceDetection | 52.64 | **54.65** | 52.01 | 52.33 | 54.60 | 52.38 | 52.38 | 52.89 | 52.44 | 50.06 | 53.06 | 52.27 |
| FingerMovements | 59.00 | 47.00 | 50.00 | 54.00 | 52.00 | **63.00** | 60.00 | 58.00 | 52.00 | 48.00 | 51.00 | 42.00 |
| HandMovementDirection | 35.14 | 29.73 | 32.43 | 22.97 | 32.43 | 28.38 | **41.89** | 32.43 | 36.49 | 27.03 | 27.03 | 33.78 |
| Handwriting | 51.88 | 48.82 | 24.94 | 52.47 | 49.65 | 53.65 | 52.59 | 50.71 | **58.24** | 5.65 | 3.29 | 34.94 |
| Heartbeat | 74.63 | 72.20 | 70.73 | 71.71 | 74.63 | 73.17 | 69.27 | 70.73 | **75.12** | 72.20 | 73.17 | 72.68 |
| JapaneseVowels | **97.57** | 96.49 | 95.41 | 96.76 | 96.76 | **97.57** | 96.76 | 97.03 | 95.41 | 12.70 | 94.32 | 97.30 |
| LSST | 50.93 | 53.45 | 51.62 | 50.57 | 51.34 | 49.92 | 50.12 | 49.76 | 50.53 | 31.39 | **54.66** | 51.42 |
| Libras | 85.00 | 78.89 | 72.78 | 84.44 | 79.44 | **86.67** | 78.33 | 77.78 | 82.22 | 38.33 | 24.44 | 62.22 |
| MotorImagery | 56.00 | 47.00 | 55.00 | 51.00 | 53.00 | **57.00** | 50.00 | 53.00 | 52.00 | 51.00 | 48.00 | 50.00 |
| NATOPS | 90.56 | **92.78** | 83.89 | **92.78** | 88.33 | 91.11 | 88.89 | 85.56 | 91.11 | 28.89 | 70.00 | 83.33 |
| PEMS-SF | 75.72 | 74.57 | 74.57 | 76.88 | 73.99 | 78.03 | 76.88 | 76.30 | 82.08 | 26.59 | **91.91** | 71.68 |
| PenDigits | 97.66 | 97.20 | **97.88** | 97.66 | 97.06 | 97.03 | 97.77 | 97.43 | 96.88 | 29.93 | 29.42 | 96.86 |
| PhonemeSpectra | 23.62 | 24.43 | 23.65 | 24.04 | **26.33** | 23.50 | 25.26 | 24.10 | 24.46 | 4.83 | 4.35 | 20.28 |
| RacketSports | **84.21** | 80.92 | 78.29 | 83.55 | 83.55 | 79.61 | **84.21** | 81.58 | 82.24 | 36.84 | 69.74 | 80.26 |
| SelfRegulationSCP1 | 84.30 | **87.03** | 82.59 | 85.32 | 86.69 | 86.01 | 79.52 | **87.03** | 80.89 | 49.83 | 76.45 | 82.59 |
| SelfRegulationSCP2 | 51.11 | **55.56** | 55.00 | 51.67 | 52.22 | 48.89 | 46.67 | 46.67 | 49.44 | 51.11 | 49.44 | 55.00 |
| SpokenArabicDigits | 96.86 | **97.73** | 97.54 | 97.32 | 97.04 | 97.27 | 97.45 | 97.00 | 97.09 | 14.19 | 71.26 | 93.72 |
| StandWalkJump | 46.67 | 26.67 | 33.33 | **53.33** | 40.00 | 40.00 | 46.67 | 46.67 | 40.00 | 26.67 | 40.00 | 46.67 |
| UWaveGestureLibrary | 82.50 | **88.12** | 75.31 | 81.25 | 80.62 | 80.00 | 79.38 | 84.06 | 71.25 | 30.00 | 14.69 | 56.88 |
| Average | **71.24** | 70.02 | 66.95 | 70.12 | 70.23 | 70.38 | 69.87 | 69.49 | 69.17 | 34.43 | 47.55 | 66.87 |

*Table 17.* 124 UCR Performance across datasets. Best per dataset in bold, second best underlined.

| Dataset | Di-COT | CoST | SimMTM | Soft | TF-C | TNC | TS2Vec | TS-TCC | CaTT | MiniRocket | Random Forest | InfoTS |
|---|---|---|---|---|---|---|---|---|---|---|---|---|
| ACSF1 | **88.00** | 85.00 | 80.00 | 84.00 | 87.00 | 84.00 | 81.00 | 81.00 | 81.00 | 59.00 | 9.00 | 78.00 |
| Adiac | **81.33** | 81.07 | 64.96 | 77.49 | 79.80 | 80.56 | 78.01 | 78.01 | 78.01 | 55.75 | 2.05 | 65.98 |
| AllGestureWiimoteX | **74.43** | 71.57 | 63.14 | 72.29 | 68.29 | 72.43 | 68.71 | 70.29 | 62.14 | 17.86 | 15.57 | 46.57 |
| AllGestureWiimoteY | **74.86** | 73.00 | 62.57 | 74.14 | 71.57 | 70.00 | 70.86 | 70.57 | 65.86 | 25.57 | 17.14 | 53.00 |
| AllGestureWiimoteZ | **73.71** | 71.86 | 61.57 | 69.43 | 68.14 | 73.43 | 69.00 | 69.86 | 61.57 | 14.71 | 13.43 | 55.14 |
| ArrowHead | 78.86 | 78.86 | 48.57 | 79.43 | 76.00 | 82.29 | 79.43 | 80.57 | 75.43 | **84.00** | 32.00 | 70.86 |
| BME | 98.67 | 96.67 | 80.00 | 98.00 | 96.67 | **99.33** | 94.00 | 89.33 | 94.00 | 97.33 | 48.00 | 73.33 |
| Beef | 60.00 | 63.33 | 56.67 | 76.67 | 70.00 | 63.33 | 60.00 | 70.00 | 66.67 | 66.67 | 30.00 | 60.00 |
| BeetleFly | **100.00** | 80.00 | 75.00 | 90.00 | 95.00 | 85.00 | 90.00 | 85.00 | 75.00 | 95.00 | 45.00 | 75.00 |
| BirdChicken | 85.00 | **100.00** | 90.00 | 90.00 | 90.00 | **100.00** | **100.00** | 90.00 | 70.00 | 90.00 | 45.00 | 75.00 |
| CBF | 99.44 | 98.67 | **100.00** | 98.67 | 98.67 | 99.33 | 98.89 | 99.56 | 99.11 | 96.67 | 33.89 | 98.44 |
| Car | **90.00** | 86.67 | 63.33 | 78.33 | 85.00 | 76.67 | 75.00 | 80.00 | 88.33 | 78.33 | 30.00 | 51.67 |
| Chinatown | 97.67 | 96.50 | 93.00 | 97.38 | 96.79 | 97.96 | 97.67 | 97.08 | 93.88 | 88.63 | 60.64 | 93.88 |
| ChlorineConcentration | **67.66** | 67.55 | 58.05 | 66.43 | 66.28 | 67.37 | 67.16 | 61.09 | 63.10 | 55.68 | 35.52 | 56.85 |
| CinCECGTorso | 67.61 | 60.07 | 42.83 | 65.87 | 62.46 | 59.78 | 64.06 | 49.86 | 62.83 | **68.26** | 26.09 | 51.16 |
| Coffee | 96.43 | 100.00 | 100.00 | 100.00 | 100.00 | 100.00 | 92.86 | 100.00 | 100.00 | 96.43 | 46.43 | 82.14 |
| Computers | **79.60** | 74.40 | 67.20 | 75.20 | 73.60 | 72.00 | 71.20 | 72.00 | 73.60 | 55.60 | 44.40 | 70.80 |
| CricketX | **77.18** | 75.38 | 57.69 | 72.82 | 74.36 | 72.82 | 71.54 | 72.56 | 67.18 | 53.59 | 10.00 | 66.15 |
| CricketY | 73.59 | **75.13** | 55.38 | 73.85 | 70.77 | 72.56 | 67.69 | 68.21 | 67.18 | 63.85 | 11.28 | 60.00 |
| CricketZ | **79.23** | 77.18 | 61.54 | 72.82 | 75.13 | 77.95 | 71.54 | 73.85 | 72.82 | 58.21 | 9.23 | 64.62 |
| Crop | 72.74 | 71.96 | **73.01** | 72.48 | 72.41 | 72.61 | 72.46 | 72.53 | 72.62 | 43.02 | 27.70 | 60.63 |
| DiatomSizeReduction | 84.97 | 86.60 | 90.85 | 85.62 | 87.91 | 88.24 | 85.29 | 88.24 | 88.56 | 81.37 | 31.70 | 83.99 |
| DistalPhalanxOutlineAgeGroup | 68.35 | 71.94 | 71.22 | 71.94 | 73.38 | 66.19 | 74.82 | 72.66 | 71.22 | 46.76 | 46.76 | 71.94 |
| DistalPhalanxOutlineCorrect | 73.19 | 75.72 | 77.17 | 78.99 | 77.17 | 78.26 | 76.81 | 75.72 | 76.09 | 67.39 | 46.74 | 76.09 |
| DistalPhalanxTW | 65.47 | 69.06 | 69.06 | 64.75 | 69.06 | 66.91 | 66.91 | 67.63 | 66.19 | 53.24 | 27.34 | 65.47 |
| ECG200 | 89.00 | 90.00 | 80.00 | 89.00 | 87.00 | 85.00 | 88.00 | 88.00 | 88.00 | 74.00 | 61.00 | 85.00 |
| ECG5000 | 93.96 | 93.96 | 93.80 | 93.53 | 94.16 | 93.91 | 93.62 | 93.89 | 94.31 | 61.84 | 23.87 | 93.58 |
| ECGFiveDays | 98.37 | 99.42 | 62.49 | 99.30 | 99.19 | 97.33 | 98.49 | 87.46 | 95.47 | 100.00 | 50.52 | 87.57 |
| EOGHorizontalSignal | **62.71** | 56.35 | 47.51 | 52.49 | 51.10 | 56.91 | 51.38 | 53.87 | 53.31 | 53.59 | 17.96 | 46.96 |
| EOGVerticalSignal | 46.69 | 46.13 | 27.90 | 40.33 | 45.58 | 43.09 | 33.98 | 44.20 | 36.46 | 53.04 | 10.50 | 39.23 |
| Earthquakes | 74.10 | 73.38 | 74.82 | 72.66 | 74.82 | 75.54 | 78.42 | 71.94 | 76.98 | 74.82 | 62.59 | 74.10 |
| ElectricDevices | 66.40 | 66.50 | 66.62 | 66.31 | 67.63 | 64.91 | 66.79 | 66.42 | 66.52 | 50.16 | 15.46 | **67.76** |
| EthanolLevel | 61.00 | 63.60 | 47.40 | 57.20 | 57.00 | 64.00 | 52.80 | 60.20 | 62.00 | 43.00 | 27.20 | 52.20 |
| FaceAll | 90.24 | 91.12 | 84.85 | 89.64 | 90.00 | 89.59 | 83.91 | 89.64 | 93.02 | 30.24 | 6.33 | 87.93 |
| FaceFour | **94.32** | 86.36 | 79.55 | 89.77 | 86.36 | 94.32 | 87.50 | 89.77 | 71.59 | 82.95 | 18.18 | 79.55 |
| FacesUCR | 92.73 | 91.66 | 78.44 | 91.71 | 90.63 | 91.07 | 92.98 | 90.54 | 86.24 | 70.59 | 6.49 | 82.54 |
| FiftyWords | 70.99 | 76.70 | 56.70 | 74.29 | 71.65 | 71.65 | 70.33 | 70.99 | 63.30 | 51.65 | 1.98 | 49.67 |
| Fish | **98.29** | 97.71 | 81.71 | 95.43 | 96.00 | 96.00 | 94.86 | 93.71 | 96.00 | 70.29 | 9.71 | 81.14 |
| FordA | 94.02 | 93.71 | 94.09 | 93.18 | 93.26 | 94.24 | 93.79 | 93.79 | 94.09 | 91.36 | 48.86 | 93.11 |
| FordB | 79.51 | 79.14 | 81.48 | 79.88 | 78.15 | 78.02 | 80.12 | 80.62 | 80.12 | 76.79 | 49.88 | 80.99 |
| FreezerRegularTrain | 98.39 | 98.74 | 87.33 | 97.96 | 97.75 | 98.88 | 97.86 | 94.91 | 98.91 | 71.96 | 50.56 | 93.89 |
| FreezerSmallTrain | 90.77 | 80.98 | 73.44 | 81.16 | 67.79 | 90.81 | 74.95 | 74.00 | 97.51 | 69.16 | 50.95 | 82.14 |
| Fungi | **95.16** | 91.94 | 83.87 | 92.47 | 95.16 | 95.16 | 89.25 | 81.18 | 90.86 | 87.10 | 4.30 | 81.18 |
| GestureMidAirD1 | 63.08 | 67.69 | 56.92 | 59.23 | 57.69 | 63.85 | 61.54 | 57.69 | 69.23 | 50.77 | 10.00 | 60.77 |
| GestureMidAirD2 | 56.92 | 62.31 | 52.31 | 50.00 | 49.23 | 55.38 | 47.69 | 53.85 | 60.77 | 46.15 | 8.46 | 47.69 |
| GestureMidAirD3 | 26.15 | 25.38 | 23.08 | 26.15 | 26.92 | 25.38 | 23.85 | 20.00 | 31.54 | 33.08 | 5.38 | 30.00 |
| GesturePebbleZ1 | 87.79 | 87.21 | 80.81 | 87.21 | 85.47 | 82.56 | 86.63 | 87.79 | 87.79 | 89.53 | 42.44 | 81.40 |
| GesturePebbleZ2 | 82.91 | 84.81 | 84.18 | 84.81 | 82.28 | 83.54 | 82.91 | 82.91 | 77.85 | 82.91 | 39.87 | 81.01 |
| GunPoint | 98.67 | 98.00 | 88.00 | 98.00 | 98.00 | 99.33 | 100.00 | 96.00 | 97.33 | 93.33 | 46.67 | 95.33 |
| GunPointAgeSpan | 99.05 | 98.10 | 95.89 | 97.78 | 99.05 | 97.78 | 94.62 | 98.10 | 97.47 | 88.29 | 82.91 | 76.58 |
| GunPointMaleVersusFemale | 100.00 | 98.73 | 99.05 | 99.68 | 100.00 | 100.00 | 98.42 | 99.68 | 97.78 | 98.42 | 88.92 | 73.42 |
| GunPointOldVersusYoung | 100.00 | 100.00 | 100.00 | 100.00 | 100.00 | 100.00 | 100.00 | 100.00 | 100.00 | 93.33 | 100.00 | 100.00 |
| Ham | 69.52 | 79.05 | 60.00 | 77.14 | 75.24 | 78.10 | 74.29 | 66.67 | 80.00 | 71.43 | 53.33 | 70.48 |
| HandOutlines | 93.78 | 91.35 | 82.70 | 90.27 | 88.38 | 91.62 | 86.49 | 90.81 | 91.89 | 92.70 | 55.14 | 77.84 |
| Haptics | 50.00 | 49.35 | 42.86 | 47.08 | 47.73 | 49.03 | 46.75 | 47.08 | 45.45 | 50.97 | 19.16 | 40.26 |
| Herring | 64.06 | 64.06 | 50.00 | 67.19 | 59.38 | 59.38 | 56.25 | 60.94 | 65.62 | 59.38 | 54.69 | 62.50 |
| HouseTwenty | 93.28 | 85.71 | 74.79 | 85.71 | 86.55 | 90.76 | 85.71 | 94.12 | 98.32 | 96.64 | 43.70 | 91.60 |
| InlineSkate | 38.73 | 27.82 | 23.27 | 29.45 | 28.00 | 33.64 | 26.18 | 24.55 | 34.00 | 35.64 | 15.45 | 25.27 |
| InsectEPGRegularTrain | 100.00 | 100.00 | 100.00 | 100.00 | 100.00 | 100.00 | 100.00 | 100.00 | 99.60 | 100.00 | 100.00 | 100.00 |
| InsectEPGSmallTrain | 100.00 | 100.00 | 100.00 | 100.00 | 100.00 | 100.00 | 100.00 | 100.00 | 100.00 | 100.00 | 100.00 | 100.00 |
| InsectWingbeatSound | 57.53 | 58.38 | 47.17 | 58.84 | 56.21 | 56.62 | 57.12 | 58.64 | 55.35 | 63.33 | 9.60 | 40.96 |
| ItalyPowerDemand | 94.17 | 96.99 | 94.95 | 96.02 | 96.11 | 95.53 | 96.40 | 97.08 | 95.63 | 95.24 | 54.52 | 95.63 |
| LargeKitchenAppliances | 87.20 | 86.40 | 68.80 | 90.93 | 86.93 | 89.07 | 83.47 | 85.87 | 87.47 | 37.33 | 38.93 | 85.07 |
| Lightning2 | 81.97 | 78.69 | 65.57 | 72.13 | 80.33 | 67.21 | 68.85 | 73.77 | 77.05 | 72.13 | 45.90 | 77.05 |
| Lightning7 | 84.93 | 76.71 | 57.53 | 73.97 | 82.19 | 75.34 | 78.08 | 78.08 | 76.71 | 68.49 | 23.29 | 61.64 |
| Mallat | 94.97 | 91.94 | 81.79 | 94.03 | 92.24 | 95.61 | 92.62 | 92.75 | 91.22 | 86.27 | 13.73 | 85.71 |
| Meat | 100.00 | 93.33 | 88.33 | 93.33 | 88.33 | 90.00 | 90.00 | 91.67 | 93.33 | 96.67 | 33.33 | 93.33 |
| MedicalImages | 77.76 | 77.24 | 71.32 | 78.55 | 78.68 | 79.21 | 78.42 | 77.24 | 78.03 | 57.89 | 12.24 | 67.89 |
| MiddlePhalanxOutlineAgeGroup | 55.19 | 51.30 | 53.90 | 53.25 | 52.60 | 51.30 | 54.55 | 55.19 | 53.25 | 18.83 | 36.36 | 54.55 |
| MiddlePhalanxOutlineCorrect | 82.82 | 77.32 | 84.88 | 80.76 | 80.76 | 80.76 | 82.47 | 81.79 | 79.73 | 77.66 | 43.30 | 80.07 |
| MiddlePhalanxTW | 49.35 | 53.90 | 56.49 | 54.55 | 53.25 | 55.84 | 55.84 | 57.79 | 56.49 | 58.44 | 33.12 | 55.84 |
| MixedShapesRegularTrain | 94.35 | 93.24 | 84.33 | 88.37 | 89.77 | 91.55 | 95.35 | 89.24 | 87.09 | 34.10 | 23.26 | 70.10 |
| MixedShapesSmallTrain | 85.20 | 83.75 | 72.91 | 78.85 | 80.82 | 83.51 | 73.61 | 82.23 | 76.16 | 55.63 | 21.65 | 69.53 |
| MoteStrain | 81.71 | 87.06 | 89.14 | 88.34 | 84.27 | 84.66 | 90.42 | 85.14 | 85.06 | 90.02 | 49.92 | 85.94 |
| NonInvasiveFetalECGThorax1 | 92.52 | 92.93 | 86.41 | 92.01 | 91.96 | 93.69 | 90.99 | 92.47 | 91.60 | 86.21 | 2.49 | 82.65 |
| NonInvasiveFetalECGThorax2 | 93.94 | 94.30 | 88.60 | 93.28 | 94.25 | 94.50 | 91.54 | 93.59 | 93.89 | 87.23 | 3.05 | 87.28 |
| OSULeaf | **92.56** | 87.60 | 76.86 | 84.71 | 85.92 | 87.19 | 80.17 | 83.88 | 78.93 | 59.09 | 17.36 | 70.66 |
| OliveOil | 86.67 | 83.33 | 76.67 | 83.33 | 80.00 | 80.00 | 83.33 | 76.67 | 80.00 | 86.67 | 33.33 | 80.00 |
| PLAID | 52.89 | 46.18 | 45.62 | 45.44 | 47.49 | 47.11 | 40.97 | 44.69 | 37.99 | 33.71 | 40.04 | 49.72 |
| PhalangesOutlinesCorrect | 79.60 | 81.00 | 75.06 | 82.40 | 81.24 | 81.00 | 80.19 | 79.95 | 79.25 | 69.58 | 47.09 | 75.06 |
| Phoneme | **29.64** | 28.48 | 26.11 | 28.32 | 28.43 | 28.53 | 27.37 | 26.85 | 21.57 | 5.06 | 2.64 | 23.36 |
| PickupGestureWiimoteZ | 74.00 | 72.00 | 80.00 | 68.00 | 78.00 | 82.00 | 72.00 | 86.00 | 78.00 | 74.00 | 32.00 | 64.00 |

**Learning Robust Temporal Features without Augmentation**

| Dataset | Di-COT | CoST | SimMTM | Soft | TF-C | TNC | TS2Vec | TS-TCC | CaTT | MiniRocket | Random Forest | InfoTS |
|---|---|---|---|---|---|---|---|---|---|---|---|---|
| PigAirwayPressure | **52.40** | 41.35 | _50.48_ | 36.54 | 41.83 | 34.13 | 40.87 | 43.75 | 29.81 | 15.87 | 12.50 | 35.10 |
| PigArtPressure | _97.12_ | 91.83 | 64.42 | 70.67 | **97.60** | 85.10 | 58.17 | _97.12_ | 44.23 | 35.58 | 19.23 | 13.46 |
| PigCVP | **94.23** | _89.42_ | 64.42 | 80.29 | 88.46 | 73.08 | 77.88 | 83.65 | 46.63 | 23.08 | 18.75 | 56.25 |
| Plane | **100.00** | **100.00** | 96.19 | **100.00** | **100.00** | **100.00** | 97.14 | 96.19 | 99.05 | 99.05 | 15.24 | 98.10 |
| PowerCons | 93.89 | 94.44 | 94.44 | 96.11 | **97.22** | 95.00 | _96.67_ | 96.11 | 92.78 | 91.67 | 90.00 | 90.00 |
| ProximalPhalanxOutlineAgeGroup | **85.85** | 84.88 | **85.85** | 81.95 | 83.90 | 81.46 | 83.41 | 82.44 | 82.93 | 61.46 | 50.24 | 82.93 |
| ProximalPhalanxOutlineCorrect | 86.94 | _89.00_ | 83.85 | 88.32 | **90.03** | 88.66 | 87.29 | 88.32 | 86.94 | 79.38 | 57.04 | 84.88 |
| ProximalPhalanxTW | 78.54 | 77.56 | **80.00** | 77.56 | 77.07 | 76.10 | 75.12 | 79.02 | _79.51_ | 75.61 | 33.66 | 78.54 |
| RefrigerationDevices | 50.67 | 50.40 | 50.40 | 48.27 | 51.20 | 53.07 | 54.67 | 53.87 | **57.60** | 33.33 | 32.80 | _55.73_ |
| Rock | 56.00 | 54.00 | 42.00 | 40.00 | 34.00 | 52.00 | 36.00 | **60.00** | _58.00_ | 40.00 | 24.00 | 50.00 |
| ScreenType | **53.33** | 48.27 | 39.47 | _49.33_ | 46.40 | 48.27 | 45.07 | 44.53 | 41.07 | 41.07 | 34.13 | 46.67 |
| SemgHandGenderCh2 | _85.17_ | 83.83 | 84.83 | 80.17 | 84.33 | **85.50** | 78.17 | _85.17_ | 81.50 | 47.17 | 57.00 | _85.17_ |
| SemgHandMovementCh2 | 58.89 | 53.11 | 56.89 | **59.56** | 59.33 | 59.11 | 56.22 | **59.56** | 42.22 | 28.22 | 33.11 | 50.89 |
| SemgHandSubjectCh2 | _71.56_ | 64.67 | 60.89 | 64.44 | 65.33 | 69.78 | 61.11 | **72.44** | 57.33 | 29.56 | 30.89 | 57.11 |
| ShakeGestureWiimoteZ | _92.00_ | 88.00 | _92.00_ | 90.00 | 88.00 | **94.00** | _92.00_ | _92.00_ | 84.00 | 74.00 | 38.00 | 86.00 |
| ShapeletSim | 92.78 | 90.56 | 98.33 | 91.67 | 93.89 | **100.00** | 97.78 | **100.00** | **100.00** | **100.00** | 45.00 | 92.78 |
| ShapesAll | _87.50_ | **88.17** | 76.00 | 83.83 | 84.17 | 86.17 | 80.00 | 84.50 | 84.33 | 33.50 | 1.67 | 63.33 |
| SmallKitchenAppliances | **84.27** | 80.00 | 75.47 | _83.20_ | 78.40 | 82.67 | 82.93 | 81.60 | 77.87 | 56.80 | 36.27 | 73.07 |
| SmoothSubspace | _96.00_ | 92.67 | 92.00 | 95.33 | 92.67 | _96.00_ | **96.67** | 93.33 | 92.67 | 83.33 | 61.33 | _96.00_ |
| SonyAIBORobotSurface1 | 92.68 | 84.53 | 83.03 | 89.18 | 83.53 | 82.86 | 80.37 | **96.51** | 92.01 | 78.37 | 50.25 | 89.18 |
| SonyAIBORobotSurface2 | 89.40 | 86.88 | 79.64 | _90.24_ | 90.14 | **91.08** | 89.40 | 89.93 | 83.84 | 86.67 | 53.31 | 87.20 |
| StarLightCurves | **97.01** | _96.98_ | 94.55 | 95.94 | 96.32 | 96.60 | 95.73 | 96.08 | 96.45 | 96.45 | 42.59 | 89.49 |
| Strawberry | **96.49** | 95.41 | 81.89 | 95.41 | 95.68 | _96.22_ | 94.05 | 93.78 | 95.95 | 66.22 | 47.30 | 91.35 |
| SwedishLeaf | _94.56_ | **94.72** | 91.04 | _94.56_ | 93.92 | 93.12 | 93.60 | 94.08 | 93.92 | 77.44 | 10.08 | 84.48 |
| Symbols | **95.08** | 86.03 | _93.57_ | 82.61 | 84.82 | 92.66 | 73.27 | 93.47 | 84.02 | 53.37 | 18.99 | 66.93 |
| SyntheticControl | **98.67** | 97.33 | _98.00_ | 97.33 | _98.00_ | _98.00_ | _98.00_ | _98.00_ | _98.00_ | 97.33 | 18.00 | 97.00 |
| ToeSegmentation1 | 94.30 | **97.37** | 95.18 | 90.35 | _96.05_ | 88.60 | 87.28 | _96.05_ | 88.16 | 95.61 | 51.32 | 86.84 |
| ToeSegmentation2 | 89.23 | 87.69 | 81.54 | 86.15 | 87.69 | 68.46 | 87.69 | **92.31** | _90.00_ | 75.38 | 54.62 | 76.92 |
| Trace | **100.00** | **100.00** | 73.00 | 98.00 | **100.00** | **100.00** | 98.00 | 96.00 | **100.00** | **100.00** | 26.00 | 98.00 |
| TwoLeadECG | _97.54_ | 93.68 | 69.01 | 87.53 | 85.16 | 97.10 | 89.29 | 86.22 | **98.77** | 49.78 | 96.58 |
| TwoPatterns | 95.30 | 96.97 | _99.90_ | 98.38 | 98.95 | 98.00 | 99.20 | 99.45 | **99.97** | 98.53 | 24.00 | 96.15 |
| UMD | **99.31** | _98.61_ | 80.56 | 97.92 | 96.53 | 97.92 | _98.61_ | 96.53 | 95.83 | 96.53 | 56.25 | 75.69 |
| UWaveGestureLibraryAll | _80.65_ | 79.79 | 72.59 | 78.39 | 77.67 | 80.07 | 76.88 | 77.92 | 73.39 | **92.16** | 15.52 | 60.19 |
| UWaveGestureLibraryX | 74.87 | _76.47_ | 74.12 | 75.82 | 75.10 | 75.32 | 75.52 | 75.24 | 75.57 | **80.40** | 16.44 | 65.49 |
| UWaveGestureLibraryY | 64.41 | _66.69_ | 62.73 | 64.46 | 62.95 | 65.16 | 63.26 | 64.80 | 64.32 | **71.47** | 14.82 | 52.01 |
| UWaveGestureLibraryZ | 72.33 | **73.14** | 71.02 | 71.22 | 71.50 | 72.17 | 72.14 | 72.19 | 72.22 | _72.92_ | 15.86 | 62.70 |
| Wafer | 99.25 | _99.29_ | 97.55 | 99.08 | 99.21 | **99.32** | 99.14 | 98.65 | 99.19 | 98.39 | 72.37 | 96.46 |
| Wine | **85.19** | 81.48 | 77.78 | **85.19** | 79.63 | 81.48 | **85.19** | 79.63 | 81.48 | 72.22 | 59.26 | **85.19** |
| WordSynonyms | 61.76 | 59.72 | 46.55 | _62.07_ | 60.97 | **62.85** | 59.87 | 60.50 | 57.37 | 45.61 | 4.55 | 40.91 |
| Worms | _75.32_ | 74.03 | 71.43 | 72.73 | 72.73 | 67.53 | 71.43 | **80.52** | 59.74 | 49.35 | 33.77 | 54.55 |
| WormsTwoClass | 74.03 | 74.03 | **80.52** | 74.03 | 75.32 | 70.13 | 74.03 | _79.22_ | 77.92 | 76.62 | 58.44 | 71.43 |
| Yoga | _82.83_ | **83.87** | 71.07 | 81.30 | 81.80 | 79.20 | 75.93 | 78.73 | 71.00 | 68.60 | 49.03 | 66.90 |
| Average | **81.33** | _80.03_ | 73.27 | 79.09 | 79.19 | 79.63 | 77.74 | 79.15 | 77.42 | 67.89 | 34.50 | 71.96 |

