# OpenReview forum: "Divide and Contrast: Learning Robust Temporal Features without Augmentation"
_ICML.cc/2026/Conference — ICML 2026 regular_

### Official Review · Reviewer_x9YW · 2026-03-09

**Soundness:** 3
**Presentation:** 2
**Significance:** 3
**Originality:** 3
**Overall Recommendation:** 4
**Confidence:** 4

**Summary:**

The paper proposes a self-supervised learning methods for time-series signals. In contrast to prior works that perform contrastive loss on entire sequence, which impose high computational cost as sequence length scales, the authors divide the sequence into overlapping subwindows and contrast adjacent blocks during training. The proposed work outperform baselines across diverse time-series datasets, and shows enhanced training efficiency.

**Compliance With Llm Reviewing Policy:**

Affirmed.

**Final Justification:**

Author's responses addressed most of my concerns.

**Key Questions For Authors:**

Most of the questions match the weakness above.
1. In time-series signals, non-adjacent segments can still be semantically similar (e.g., periodic or repetitive patterns). Can authors provide more discussions on how could this adjacency-based assumption affect the learned representations, and have the authors considered simple temporal proximity?
2. Could the authors elaborate on how the k-sampling supports learning both short- and long-range temporal dependencies in practice? The granularity depends on the selection of k_max, which is << T, and in this case, how could the model learn global long-range dependencies that represents the full sequence?
3. Can the authors clarify the full downstream finetuning/inference pipeline on how the sub-block structures are used? How could the cross-block relationship be obtained in this scenario?

**Limitations:**

yes

**Strengths And Weaknesses:**

Strength: I enjoyed reading the paper. The paper identifies a new insight for contrasting along the temporal dimension for the time-series signals. The proposed contrastive objectives within dynamic slices (or subwindows) could significantly improve the computational efficiency as the sequence scales. The dynamic k can also allow the models to learn different granularities of the input. The authors evaluated the proposed work on comprehensive experiments and achieves SOTA performance across most cases.

Weakness: I have a few concerns regarding the design of the work. Time-series signals exhibit periodic events. Therefore, cross-sequence slices that act as negative pairs could contain similar information due to periodic events that are common in these applications. Past literature tried to address this by enforcing soft learning objectives as regularization, where the similarity between neighboring slices is less than or equal to the similarity between distance slices [1]. Moreover, it is unclear how the representations from different slices are fused together for downstream applications. I believe that subwindowing the sequence could lose certain temporal consistency that could be non-trivial for downstream classifications.

Minor: Some tables are too small. Overview figures can also be enhanced to provide more details on positive and negative pairs for contrasting.

Liu, S., Kimura, T., Liu, D., Wang, R., Li, J., Diggavi, S., Srivastava, M. and Abdelzaher, T., 2023. Focal: Contrastive learning for multimodal time-series sensing signals in factorized orthogonal latent space. Advances in Neural Information Processing Systems, 36, pp.47309-47338.

---

> ### Author Rebuttal · Authors · 2026-03-31
>
> Thank you for the thoughtful and positive assessment of our work. We are glad you found the core idea of within-instance dynamic sub-block contrasting insightful.
>
> > *Overview figures can also be enhanced to provide more details on positive and negative pairs for contrasting.*
>
> Thank you for this suggestion! We will improve the overview figure to more clearly illustrate anchor, positive, and negative pairs, and increase the readability of tables in the final version.
>
> ---
>
> > 1. In time-series signals, non-adjacent segments can still be semantically similar
>
> This is an important point.
>
> In Di-COT, contrastive learning is performed within a single instance, where sub-blocks are sampled from the same window. This differs from cross-instance contrastive setups and ensures that all segments share a common semantic context.
>
> We agree that periodic patterns may lead to non-adjacent segments being semantically similar. Of course identifying and leveraging these similarities would require labeling, which is important to avoid in our context. Fortunately, our objective does not enforce global dissimilarity across distant segments; rather, it imposes a local consistency prior by encouraging adjacent sub-blocks to have similar representations.
>
> Importantly, the rolling nature of our adjacent contrast, where block $i$ serves as anchor with block $i+1$ as positive, then block $i+1$ becomes the anchor with block $i+2$ as positive, and so on, creates a chain of local consistency constraints that can propagate transitively through the entire sequence.
>
> Additionally, the use of overlapping sub-blocks and stochastic partitioning exposes the model to multiple shifted views of the same signal, implicitly promoting shift/translation invariance, which is particularly beneficial for periodic data.
>
> Empirically, strong performance across diverse datasets with varying temporal dynamics (UCR and UEA) suggests that this local consistency assumption is robust in practice.
>
> ---
>
> > 2. Could the authors elaborate on how the k-sampling supports learning both short- and long-range temporal dependencies in practice?
>
> While each sub-block has length $L < T$, the model is not restricted to local information for two reasons:
>
> - Overlapping sub-blocks ensure that information is shared across neighboring segments, allowing dependencies to propagate across the full sequence.
> - Stochastic $k$-sampling across training iterations exposes the model to different partitions of the same sequence, effectively covering diverse temporal contexts and granularities over time.
>
> Specifically, coarse partitions with small $k$ capture long-range structure, while fine partitions with large $k$ capture local patterns, allowing the model to learn representations that are robust to varying temporal dynamics, including periodic and seasonal structure.
>
> As a result, although supervision is applied locally, the encoder learns representations that integrate information across the entire sequence. Importantly, during inference, the encoder processes the full instance (sequence length $T$), enabling it to harness global temporal dependencies learned during training.
>
> ---
>
> > 3. Can the authors clarify the full downstream finetuning/inference pipeline on how the sub-block structures are used? How could the cross-block relationship be obtained in this scenario?
>
> During training, our method applies stochastic sub-block partitioning to construct contrastive pairs. However, this sampling strategy is not used during inference.
>
> At evaluation time, the encoder processes the full input instance (sequence length, $T$) to produce a single representation with dimension $F$, which is then used for downstream tasks (e.g., linear probing, kNN, clustering).
>
> Thus, sub-block partitioning is purely a training-time supervision mechanism, and the learned encoder naturally captures cross-block relationships when applied to the full sequence.
>
> ---
>
> Again, we appreciate the questions and hope our response will help foster understanding of our work.

---

> > ### Author Rebuttal · Reviewer_x9YW · 2026-04-03
> >
> > Thank you for your detailed rebuttal responses and clarifications.

---

### Official Review · Reviewer_i34q · 2026-03-09

**Soundness:** 3
**Presentation:** 3
**Significance:** 2
**Originality:** 2
**Overall Recommendation:** 4
**Confidence:** 3

**Summary:**

This paper proposes Di-COT, a self-supervised method for time-series representation learning based on contrasting overlapping sub-blocks within each sequence. By removing augmentation and multiple encoder passes, and by reformulating temporal contrastive learning as a cross-entropy classification objective, the method aims to improve both efficiency and robustness. Experiments across large-scale datasets and UCR/UEA show strong accuracy, faster training, and good performance in low-label and transfer scenarios.

**Compliance With Llm Reviewing Policy:**

Affirmed.

**Final Justification:**

**Thank you to the authors for the detailed rebuttal and the additional experiments.**

My main concerns have been substantially addressed. I also appreciate that the authors have agreed to incorporate my suggestions into the revision, which I view positively. Therefore, I will raise the soundness score and maintain my positive overall evaluation.

That said, for the revision, the paper would be further strengthened by providing deeper analysis beyond performance improvements on Positive Selection, potentially through additional experiments or more thorough interpretation.

**Key Questions For Authors:**

1. Can you justify more clearly why only the preceding sub-block is used as the positive target, rather than the next sub-block or a bidirectional temporal neighbor?
2. Can you explain why the UCR/UEA experiments are conducted with a single fixed random seed, and why this practice is commonly followed in prior work?
3. Can you clarify why the cross-dataset transfer evaluation is restricted to a subset of aligned variables, instead of adopting a more standard transfer-learning setup as in prior work such as SoftCLT?

**Limitations:**

yes

**Strengths And Weaknesses:**

### Soundness
- The overall experimental design is reasonable, with broad evaluation across multiple settings.
- The core assumption is that adjacent sub-blocks are semantically similar, but it is not fully justified why only the **preceding** sub-block is treated as the positive target. In many time-series settings, both the previous and the next sub-block could plausibly be considered similar to the anchor. A comparison with next-only or bidirectional positives would strengthen the claim.
- For the UCR and UEA benchmarks, the paper uses only a **single fixed random seed**.
- In Section 4.5, the cross-dataset transfer setup aligns only a subset of variables across datasets. While this is a controlled setting, it raises questions about how representative the transfer results are. The paper should better explain why a more standard transfer-learning protocol, such as those used in prior work like SoftCLT([1]), was not adopted.
- The evaluation is largely limited to **time-series classification**, which makes the scope of the SSL contribution somewhat narrow.

### Presentation
- The paper is generally well written and easy to follow.
- In Section 4.2 (Low-Label Regime), the text discusses **Random Forest** and **MiniRocket**, but their results are not included in Table 3. This makes the presentation slightly inconsistent.

### Significance
- The problem is important: self-supervised learning for time-series representation aims to reduce reliance on labeled data while maintaining strong downstream performance.
- The paper also addresses a meaningful practical issue, since many existing approaches are computationally expensive or rely on assumptions that do not hold across diverse temporal dynamics.

### Originality
- The main novelty lies in replacing timestep-level contrast with **sub-block-level contrast**, together with overlapping windows and a cross-entropy-based temporal contrastive formulation. This combination is meaningful and practically useful.
- That said, the originality appears **moderate rather than high**. Sub-block-based design is already common in time-series modeling, and interpreting contrastive learning through a **cross-entropy classification** view has also appeared in prior work([1],[2]). Thus, the contribution is better characterized as a strong and well-executed refinement of existing ideas rather than a fundamentally new paradigm.

[1] Lee, S., Park, T., & Lee, K. (2023). Soft contrastive learning for time series. arXiv preprint arXiv:2312.16424.

[2] Lee, Kibok, et al. "i-mix: A domain-agnostic strategy for contrastive representation learning." *arXiv preprint arXiv:2010.08887* (2020).

---

> ### Author Rebuttal · Authors · 2026-03-31
>
> We are grateful for the constructive feedback and positive assessment of our experiments. We are glad you found our paper to be well written and easy to follow. Thank you.
>
> ---
>
> > *"In Section 4.2 (Low-Label Regime), the text discusses Random Forest and MiniRocket, but their results are not included in Table 3. This makes the presentation slightly inconsistent."*
>
> Thank you for pointing this out. We will correct this inconsistency by including the corresponding results in Table 3 in the final version.
>
> | Method          | ECG              | HARTH            | PAMAP2                     | SKODA            | SLEEP            | WISDM2           | Avg. Acc ± 95% CI |
> |-----------------|------------------|------------------|----------------------------|------------------|------------------|------------------|------------------|
> | Raw Data        | 51.67 ± 0.00     | 46.23 ± 0.00     | 57.17 ± 0.00               | 94.54 ± 0.00     | 35.58 ± 0.00     | 40.73 ± 0.00     | 54.32 ± 0.00     |
> | Random Forest   | 52.50 ± 0.00     | 84.59 ± 0.00     | 64.94 ± 0.00          | 92.58 ± 0.00     | 25.57 ± 0.00     | 41.98 ± 0.00     | 60.36 ± 0.00     |
> | MiniRocket      | 62.06 ± 7.10     | 19.86 ± 4.52     | 5.40 ± 1.21                | 17.87 ± 6.28     | 44.63 ± 0.54     | 40.38 ± 0.36     | 31.47 ± 2.75     |
>
> ---
>
> > 1. Can you justify more clearly why only the preceding sub-block is used as the positive target, rather than the next sub-block or a bidirectional temporal neighbor?
>
> We have explored this. As shown below, using the preceding sub-block consistently yields slightly better performance than using the next sub-block, although the gap is small (≈0.7–1% across datasets). This suggests that the method is largely robust to the choice of temporal direction.
>
> | Positive Selection                | Large     | UCR     | UEA     |
> |-------------------------------|--------|--------|--------|
> | Preceding Sub-block Contrast  | 83.07  | 81.33  | 71.24  |
> | Next Sub-block Contrast       | 82.22 (-1.03%) | 80.59 (-0.90%) | 70.70 (-0.76%) |
>
> Regarding bidirectional neighbors, incorporating multiple positives within the standard InfoNCE formulation would require either modifying the objective or computing multiple contrastive loss terms, which increases computational cost. Since a key goal of our method is efficiency, we adopt the simplest effective design using a single preceding sub-block.
>
> ---
>
> > 2. Can you explain why the UCR/UEA experiments are conducted with a single fixed random seed, and why this practice is commonly followed in prior work?
>
> This is standard practice in the time-series SSL literature ([1],[2],[3]), due to the large number of datasets (128 UCR, 30 UEA) and the high computational cost of running multiple seeds across all methods.
>
> To ensure statistical reliability, we complement the UCR/UEA evaluation with multiple-seed experiments (5 runs) on six large-scale real-world datasets (>20k samples each), where we report mean and standard deviation across all methods.
>
> ---
>
> > 3. Can you clarify why the cross-dataset transfer evaluation is restricted to a subset of aligned variables, instead of adopting a more standard transfer-learning setup as in prior work such as SoftCLT?
>
> The primary challenge in cross-dataset transfer for multivariate time series is mismatched sensor dimensionality and semantics across datasets.
>
> In our setup, we restrict all datasets to a shared 3-axis accelerometer subspace (x, y, z) to ensure consistent semantic alignment without modifying the architecture during downstream feature extraction.
>
> In contrast, prior work such as SoftCLT [1] primarily considers univariate settings, where such alignment is not required. Extending to heterogeneous multivariate settings often requires architectural adaptation or channel-wise assumptions.
>
> ---
> [1] Lee, S., et al. "Soft Contrastive Learning for Time Series." ICLR, 2024.
>
> [2] Luo, D., et al. "Time Series Contrastive Learning with Information-Aware Augmentations." AAAI, 2023.
>
> [3] Yue, Z., et al. "TS2Vec: Towards Universal Representation of Time Series." AAAI, 2022.

---

> > ### Author Rebuttal · Reviewer_i34q · 2026-04-02
> >
> > **Thank you for the detailed rebuttal.**
> >
> > I appreciate the clarifications on Q2, Q3, and the updates to Table 3. These points are now resolved. However, the following issues require further evidence:
> >
> > 1. Empirical Justification for Efficiency (Q1)
> > The argument that bidirectional neighbors are avoided solely for "efficiency" lacks empirical grounding. To justify this design choice, please provide a brief quantitative analysis or ablation study comparing the performance gain versus the computational overhead of a bidirectional approach.
> >
> > 2. Scope of Contribution: Classification vs. Forecasting (Soundness 5)
> > While the paper claims to learn "Robust Temporal Features," the evaluation is limited to classification. In time-series SSL, forecasting is a crucial benchmark. To support your general claim, you should either:
> > - Include forecasting results to demonstrate generalizability, or
> > - Refine the scope of the paper (including title and abstract) to focus specifically on "Representation Learning for Classification."
> >
> > Please clarify your rationale or provide a plan for manuscript revision to align the claims with the experimental scope.

---

> > > ### Author Response · Authors · 2026-04-04
> > >
> > > **Thank you for taking the time to carefully go through our rebuttal and for providing constructive and actionable feedback.**
> > >
> > > ---
> > >
> > > > 1. Ablation study comparing the performance gain versus the computational overhead of a bidirectional approach.
> > >
> > >
> > > Under our formulation (Eq. 1), the cross-entropy objective treats each sub-block $j$ as an anchor with a single positive label $p^*(j) = j-1$, and all other $k-1$ sub-block positions as negatives. This is equivalent to a $k$-way classification problem over sub-block indices, which admits a clean and fully parallelizable matrix implementation via the similarity matrix $S^{(i)} \in \mathbb{R}^{k \times k}$.
> > >
> > > Extending this to multiple positives per anchor, specifically, treating both $j-1$ and $j+1$ as positives simultaneously breaks the single-label cross-entropy formulation, since `F.cross_entropy` expects exactly one target index per anchor.
> > >
> > > This leaves the alternative of computing multiple loss terms. Following your suggestion, we conducted additional ablation experiments to evaluate this design choice. The bidirectional approach has a slightly higher average score on some metrics, and slightly lower on others; overall, the results show no significant improvement when using bidirectional positives.
> > >
> > > | Positive Selection                | UCR     | UEA     | Linear | Linear 1% | kNN (500) | NMI  | ARI  | Transfer |
> > > |----------------------------------|---------|---------|--------|-----------|------|------|------|----------|
> > > | Preceding Sub-block     | 81.33   | 71.24   | 83.07  | 76.36     | 74.34| 0.508 | 0.406| 72.12 |
> > > | Next Sub-block          | 80.59  | 70.70 | 82.22 | 75.94 | 74.18 | 0.505 | 0.403 | 72.49 |
> > > | Bidirectional (Preceding + Next) | 81.10 | 70.38 | 82.66 | 76.42 | 74.30| 0.509 | 0.408 | 71.98 |
> > >
> > > Somewhat surprisingly, we also find that the additional computational overhead of computing two different loss terms using the next and preceding positives is not substantial. To be specific, on the large-scale benchmarks, the training time for bidirectional contrast is 10382 seconds, compared to 10374 seconds for preceding sub-block contrast.
> > >
> > > We will include this ablation table and a brief discussion in the Appendix, demonstrating that the choice of constructing positive pairs from preceding sub-blocks, next sub-blocks, or bidirectional does not meaningfully affect the quality of the learned representations across any evaluation setting.
> > >
> > >
> > >
> > > ---
> > >
> > > > 2. Scope of Contribution
> > >
> > > In addition to classification, our method achieves competitive performance in clustering and $k$-NN, both of which measure representation quality directly and do not depend on a trained classifier. Our evaluation protocols collectively assess generalization (across 158 diverse datasets and settings), label efficiency, semantic structure, and transferability, which we believe are the primary objectives of self-supervised representation learning and align with the goals outlined in our introduction.
> > >
> > > We agree that forecasting is an important task in time-series research, and will clarify in the paper that such evaluations are beyond the current scope and are left for future work.
> > >
> > > We also note that this scope is not an arbitrary restriction, but reflects a fundamental property of contrastive learning. Contrastive objectives learn discriminative representations by bringing semantically similar instances closer in the embedding space and pushing dissimilar ones apart. This inductive bias is well-suited for classification, clustering, and retrieval tasks as seen in prior works in time-series SSL using contrastive learning such as TNC (Tonekaboni et al., ICLR 2021), TS-TCC (Eldele et al., IJCAI 2021), and TF-C (Zhang et al., NeurIPS 2022).
> > >
> > > That said, we agree that the current framing can be made more precise. To address this, we will revise the abstract and limitations to better reflect the scope of the paper.
> > >
> > > Specifically, we propose the following revisions:
> > >
> > >
> > >
> > > * **Revised abstract (final sentence):**
> > >   *“Extensive experiments on six large-scale real-world datasets, as well as the UCR and UEA benchmarks, demonstrate that Di-COT learns semantically structured, transferable representations, evidenced by strong performance on classification, clustering, $k$NN, and cross-domain transfer, while achieving state-of-the-art performance with substantially reduced training time.”*
> > >
> > > * **Revised limitations:**
> > >   *“Di-COT is designed around a contrastive objective that learns discriminative representations by drawing semantically similar instances closer in the embedding space, an inductive bias well-suited for classification, clustering, and retrieval tasks, but possibly orthogonal to forecasting tasks that require fine-grained temporal prediction. Extending Di-COT to such tasks may require a fundamentally different learning objective and is left for future work.”*
> > >
> > > ---
> > >
> > > These edits will strengthen the paper; we are grateful for your time and constructive feedback.

---

### Official Review · Reviewer_HnWB · 2026-03-10

**Soundness:** 2
**Presentation:** 2
**Significance:** 2
**Originality:** 2
**Overall Recommendation:** 3
**Confidence:** 4

**Summary:**

This paper proposes Divide and Contrast (Di-COT), a self-supervised framework for time-series representation learning that eliminates the need for data augmentations. The method stochastically partitions each time series into overlapping temporal sub-blocks and performs contrastive learning by treating temporally adjacent blocks as positive pairs and non-adjacent ones as negatives. The objective is formulated as a cross-entropy classification over sub-block indices, which can be interpreted as a reparameterization of the InfoNCE loss and results in computational complexity that depends on the number of sub-blocks rather than the sequence length. Experiments on multiple large-scale datasets and the UCR/UEA archives demonstrate competitive performance with substantially reduced training time.

**Compliance With Llm Reviewing Policy:**

Affirmed.

**Final Justification:**

The rebuttal addresses several of my concerns and improves my understanding of the method. While some limitations remain, I will raise my score to Weak Reject.

**Key Questions For Authors:**

**1. Validity of the Positive/Negative Pair Definition**
* The proposed method defines temporally adjacent sub-blocks as positive pairs and all other sub-blocks within the same sequence as negatives. However, temporal adjacency does not necessarily imply semantic similarity, especially in time-series with periodic or seasonal patterns where distant segments may be more similar than adjacent ones. Could the authors provide empirical evidence or theoretical justification supporting this design choice? In particular, how robust is the method when the assumption of local temporal smoothness does not hold? Clarifying this point would help determine whether the proposed contrastive objective is generally applicable or only effective for specific types of time series.

**2. Applicability Beyond Classification Tasks**

* Most experiments in the paper evaluate the learned representations using classification-based downstream tasks. However, representation learning for time series is often motivated by applications such as forecasting, anomaly detection, or change-point detection. Can the authors provide evidence that the learned representations are also effective for these tasks? Demonstrating performance on at least one additional task would significantly strengthen the claim that the method learns generally useful temporal representations.

**3. Role of Instance-wise Contrastive Signals**

* The proposed framework relies exclusively on intra-instance contrastive learning within a single time series and does not utilize relationships across different instances. Could the authors clarify whether incorporating instance-wise contrastive signals would improve representation quality? For example, have the authors experimented with combining intra-instance and inter-instance contrastive objectives?

**4. Handling of Multivariate Dependencies**

* Although many real-world datasets are multivariate, the proposed framework mainly focuses on temporal partitioning within a sequence and provides limited discussion on how dependencies across variables are modeled. Could the authors clarify how the method captures cross-variable relationships in multivariate time series? If the approach implicitly assumes channel independence, this assumption may limit its applicability to more complex multivariate settings.

**Limitations:**

The limitations of the proposed approach are not sufficiently discussed in the paper. The empirical evaluation focuses almost exclusively on classification tasks, which provides a relatively narrow assessment for a method intended for general time-series representation learning, and it remains unclear whether the learned representations would be effective for other important problems such as forecasting, anomaly detection, or change-point detection. In addition, the proposed contrastive objective relies on the assumption that temporally adjacent sub-blocks form positive pairs while non-adjacent segments are treated as negatives. This assumption lacks clear theoretical justification and may not hold for many real-world time series, particularly those exhibiting periodic or seasonal structures where distant segments may be more semantically related than adjacent ones. In such cases, the proposed pair construction could introduce noisy supervision during training. Finally, the framework mainly emphasizes temporal partitioning within individual sequences and provides limited discussion on how multivariate dependencies or cross-instance relationships are modeled, which raises concerns about the generality of the learned representations in more complex time-series settings.

**Strengths And Weaknesses:**

**Strengths**

* The paper proposes a self-supervised framework for time-series representation learning that avoids the use of data augmentations or masking strategies, thereby reducing potential biases or semantic distortions introduced by artificially transformed samples.

* By performing contrastive learning at the sub-block level rather than at the timestep level, the method improves computational efficiency and reduces the training cost.

* The stochastic sampling of the number of sub-blocks during training naturally exposes the model to multiple temporal granularities, allowing it to capture both short-term and long-term temporal dependencies.

* The proposed divide-and-contrast strategy is conceptually simple and leverages the intrinsic temporal structure of time-series data without requiring additional architectural components or complex training procedures.

**Weaknesses**

* The current empirical evaluation focuses almost exclusively on classification tasks, which is a relatively limited scope for a general-purpose time-series representation learning method. Many important applications of time-series modeling involve forecasting, anomaly detection, or change-point detection, and the effectiveness of the proposed representation for these tasks remains unclear.

* The method assumes that temporally adjacent sub-blocks should form positive pairs, while non-adjacent blocks are treated as negative pairs. However, this design choice lacks a strong theoretical justification. In many real-world time-series datasets, temporal adjacency does not necessarily imply semantic similarity. In particular, for periodic or seasonal time-series, segments that are far apart in time may exhibit stronger similarity than adjacent segments. Under such conditions, the proposed positive/negative definition may incorrectly treat semantically similar segments as negatives, potentially introducing noisy supervision during training. This limitation is not sufficiently discussed in the paper.

* The framework relies solely on intra-instance contrastive learning within a single sequence, without leveraging relationships across different instances. It remains unclear whether the absence of instance-wise contrastive signals limits the representational power of the learned embeddings.

* Although many recent representation learning methods explicitly consider multivariate time-series structures, the proposed approach mainly focuses on temporal partitioning and provides limited discussion on how cross-variable dependencies are captured.

---

> ### Author Rebuttal · Authors · 2026-03-31
>
> Thank you for providing such detailed and valuable comments.
>
> ---
>
> > 1. Validity of the Positive/Negative Pair Definition
>
> Thank you for this important point.
>
> Our method does not assume global temporal smoothness, but rather a local semantic consistency prior within a sampled instance window. This is consistent with other time-series SSL pipelines ([2], [3], [4]), where each training instance is constructed as a window that is also implicitly assumed to be semantically coherent (e.g., sharing the same label or regime). Within this context, our approach allows a more dynamic signal than these previous works.
>
> The stochastic sampling of the number of sub-blocks $k$ during training exposes the model to multiple temporal granularities, allowing the approach to be applied to datasets that change at different rates. Coarse partitions with small $k$ capture long-range structure, while fine partitions with large $k$ capture local patterns, allowing the model to learn representations that are robust to varying temporal dynamics.
>
> We agree that for periodic or long-range dependencies, distant segments may exhibit higher similarity than adjacent ones. Our objective does not explicitly enforce long-range alignment; instead, it promotes local consistency through rolling adjacent contrast ($i \rightarrow i+1$, $i+1 \rightarrow i+2$, etc.). This induces a form of transitive structure across the sequence, which can help propagate information beyond immediate neighbors.
>
> For these reasons, our method performs well across a broad suite of datasets with diverse dynamics, including periodic and non-periodic signals (UEA & UCR archives), suggesting that our local consistency prior is sufficiently general in practice.
>
> ---
>
> > 2. Applicability Beyond Classification Tasks
>
> Our evaluation is designed to assess representation quality in the context of self-supervised learning, where the primary objective is to learn useful features from unlabeled data that remain effective under limited supervision. To this end, we adopt standard protocols that directly measure this objective: Full linear evaluation, Low-label linear evaluation, Non-parametric kNN evaluation, Clustering, Cross-domain transfer, and t-SNE visualization.
>
> These settings provide a direct measure of label efficiency, generalization, and semantic organization, which are the core goals of SSL and are consistent with prior works ([1], [2], [3]).
>
> We agree that extending the method to forecasting or anomaly detection is valuable, and will clarify in the paper that such evaluations are beyond the current scope and leave them as future work.
>
> ---
>
> > 3. Role of Instance-wise Contrastive Signals
>
> A key contribution of our work is to show that effective supervision can be derived entirely from intra-instance structure, without requiring inter-instance contrastive signals.
>
> While prior methods ([2], [3]) rely on inter-instance relationships, our results demonstrate that intra-instance sub-block contrasting alone is sufficient to learn strong representations, while also being more computationally efficient and avoiding reliance on batch composition.
>
> This suggests that inter-instance contrast is not strictly necessary, and that meaningful supervision can instead be derived from the internal structure of each sequence.
>
> It may well be that combining inter-instance and intra-instance contrastive objectives would lead to better representations, and this would make interesting future work. However, it would also increase computational cost and introduce dependence on batch composition, which is not aligned with our efficiency-focused design goal in this paper.
>
> ---
>
> > 4. Handling of Multivariate Dependencies
>
> We clarify that our method operates on both univariate and multivariate time-series inputs, and that sub-block partitioning is applied jointly across all channels. Therefore, the encoder processes the entire multivariate signal, implicitly capturing cross-variable dependencies through the shared backbone (InceptionTime).
>
> This design is consistent with several prior SSL methods ([1],[2],[3],[4]), which rely on the encoder architecture rather than explicit factorization to model cross-variable interactions.
>
> Empirically, strong performance on multivariate datasets (5/6 large-scale datasets and 28 UEA datasets) suggests that the model effectively captures such dependencies.
>
> ---
>
> Once again, thank you for the detailed comments and questions. We hope these responses have effectively addressed your concerns.
>
> ---
> [1] Tonekaboni, S., et al. "Unsupervised Representation Learning for Time Series with Temporal Neighborhood Coding." ICLR, 2021.
>
> [2] Eldele, E., et al. "Time-Series Representation Learning via Temporal and Contextual Contrasting." IJCAI, 2021.
>
> [3] Zhang, X., et al. "Self-Supervised Contrastive Pre-Training for Time Series via Time-Frequency Consistency." NeurIPS, 2022.
>
> [4] Yue, Z., et al. "TS2Vec: Towards Universal Representation of Time Series." AAAI, 2022.

---

> > ### Author Rebuttal · Reviewer_HnWB · 2026-04-04
> >
> > Thank you for the detailed rebuttal. I appreciate the clarifications and additional explanations provided.
> >
> > 1. The discussion on the positive/negative pair construction is helpful, and the connection to local consistency and prior SSL frameworks makes the design choice more reasonable. While concerns remain for cases with strong periodicity or long-range dependencies, the empirical results suggest that the approach works well in practice.
> >
> > 2. I also acknowledge that the evaluation protocols (e.g., linear evaluation, kNN, clustering, and transfer) are standard in self-supervised learning and provide meaningful evidence of representation quality, even though additional tasks could further strengthen the claim.
> >
> > 3. The justification for focusing on intra-instance contrastive learning is reasonable, particularly from an efficiency perspective, and the results demonstrate that it can learn competitive representations without relying on inter-instance signals.
> >
> > Overall, the rebuttal addresses several of my concerns and improves my understanding of the method. While some limitations remain, I will raise my score to Weak Reject.

---

> > > ### Author Response · Authors · 2026-04-04
> > >
> > > Thank you for taking the time to carefully engage with our rebuttal and for adjusting your score. We are glad the rebuttal addresses several of your concerns and improves your understanding of our method.

---

### Official Review · Reviewer_CvHD · 2026-03-15

**Soundness:** 3
**Presentation:** 3
**Significance:** 1
**Originality:** 2
**Overall Recommendation:** 4
**Confidence:** 3

**Summary:**

The paper proposes Di-COT (Divide and Contrast), an unsupervised self-supervised representation learning framework for time series. The core idea of this method is to randomly divide each time series into several overlapping sub-blocks and perform contrastive learning on adjacent sub-blocks within the same sequence to learn high-quality, transferable time series representations, without relying on traditional data augmentation or multiple encoder passes. This design allows Di-COT to preserve the semantic consistency of the sequence context, reduce training overhead, and provide dense supervision signals to enhance the robustness and generalization of the learned representations.

**Compliance With Llm Reviewing Policy:**

Affirmed.

**Key Questions For Authors:**

1.The figures you presented make it difficult to discern your key innovations. While they are rigorous, this is a significant issue. There are too few figures, which does not match the quality expected for ICML.
2.The experiments on downstream tasks are not comprehensive.
3.The originality is limited; the work mainly builds upon and modifies previous methods, representing more of an engineering improvement.

**Limitations:**

是

**Strengths And Weaknesses:**

Soundness: The technique is reasonable, the experiments are relatively complete, and the method is logically sound. However, results for some downstream tasks are missing.
Presentation: The figures are somewhat limited and do not clearly highlight the innovation, though they are rigorous.
Significance: Moderate, with limited impact on the field.
Originality: The method is an improvement based on prior work, representing an incremental rather than a fully novel contribution.

---

> ### Author Rebuttal · Authors · 2026-03-31
>
> We appreciate your valuable feedback and pleased to hear that you find our method logically sound. We hope the following responses fully address your concerns.
>
> ---
>
> > 1. The figures you presented make it difficult to discern your key innovations
>
> Thank you for taking the time to examine the figures. We agree that the Figure 2 (Di-COT framework) could better highlight the key idea of our method. We will revise Figure 2 to more clearly emphasize our core contribution, namely within-instance dynamic sub-block contrasting.
>
> ---
>
> > 2. The experiments on downstream tasks are not comprehensive
>
> Our evaluation follows standard and widely adopted protocols for assessing representation quality in time-series self-supervised learning. Specifically, we evaluate across multiple complementary settings:
>
> - Linear probing (full) on 124 UCR + 28 UEA datasets
> - Evaluation on 6 large-scale datasets (>20k instances), including:
>   - Linear probing (full and low-label regimes)
>   - kNN evaluation across multiple shot settings
>   - Clustering (NMI and ARI)
>   - Cross-domain transfer
>   - t-SNE visualization of embedding geometry
>
> These evaluations collectively assess generalization, label efficiency, semantic structure, and transferability, which are the primary objectives of self-supervised representation learning.
>
> We are open to specific suggestions of additional experiments that would add clarity; however, we believe the current set of experiments is consistent with prior self-supervised representation learning works in time-series ([1], [2],[3]).
>
> ---
>
> > 3. The work mainly builds upon and modifies previous methods
>
> We agree that our work builds on the contrastive learning framework; however, our contribution is orthogonal to prior augmentation-based designs.
>
> Specifically, we introduce dynamic within-instance sub-block contrasting, which:
>
> - Removes the need for data augmentation, masking, and multiple encoder passes
> - Provides dense supervision within a single instance through rolling adjacent contrast
> - Reduces computational overhead while preserving temporal semantics
>
> To our knowledge, contrastive learning purely within a single instance via stochastic sub-block partitioning has not been explored. This design offers a simple, efficient, and novel alternative to existing pipelines.
>
> ---
> [1] Tonekaboni, S., et al. "Unsupervised Representation Learning for Time Series with Temporal Neighborhood Coding." ICLR, 2021.
>
> [2] Eldele, E., et al. "Time-Series Representation Learning via Temporal and Contextual Contrasting." IJCAI, 2021.
>
> [3] Zhang, X., et al. "Self-Supervised Contrastive Pre-Training for Time Series via Time-Frequency Consistency." NeurIPS, 2022.

---

> > ### Author Rebuttal · Reviewer_CvHD · 2026-04-05
> >
> > The issues have been explained, and the score will not be changed. The original rating was already a weak accept, and the rebuttal did not warrant raising it to a 5.

---

### Decision · Program_Chairs · 2026-04-30

**Decision:**

Accept (regular)

**Comment:**

Given all comments by reviewers and authors, this is a borderline paper, but after weighting all responses I recommend acceptance.

The paper proposes a contrastive learning objective for training time series representations by pooling positive/negative samples across “sub blocks” along the time series.

A recurring theme in the reviews was the incremental nature of the work and the distinction to some previous papers reviewers shared. However, reviewers posivitely noted that scope of the experiments, and the empirical performance of the approach.

The depth of the empirical validation is the main reason for leaning towards acceptance. The method addresses and important problem in time series modeling and is empirically successfull and evaluated on a breadth of datasets.

A requirement until the camera ready is to improve the clarity of presentation: The paper tables are essentially unreadable in its current form, also noted by reviewers. The authors should focus on presenting the main results in the main paper, and adding full results tables in the appendix where appropriate.